# Belief-Enriched Pessimistic Q-Learning against Adversarial State Perturbations

**Xiaolin Sun**
Department of Computer Science
Tulane University
New Orleans, LA 70118
xsun12@tulane.edu

**Zizhan Zheng**
Department of Computer Science
Tulane University
New Orleans, LA 70118
zzheng3@tulane.edu

## Abstract

Reinforcement learning (RL) has achieved phenomenal success in various domains. However, its data-driven nature also introduces new vulnerabilities that can be exploited by malicious opponents. Recent work shows that a well-trained RL agent can be easily manipulated by strategically perturbing its state observations at the test stage. Existing solutions either introduce a regularization term to improve the smoothness of the trained policy against perturbations or alternatively train the agent's policy and the attacker's policy. However, the former does not provide sufficient protection against strong attacks, while the latter is computationally prohibitive for large environments. In this work, we propose a new robust RL algorithm for deriving a pessimistic policy to safeguard against an agent's uncertainty about true states. This approach is further enhanced with belief state inference and diffusion-based state purification to reduce uncertainty. Empirical results show that our approach obtains superb performance under strong attacks and has a comparable training overhead with regularization-based methods. Our code is available at https://github.com/SliencerX/Belief-enriched-robust-Q-learning.

## 1 Introduction

As one of the major paradigms for data-driven control, reinforcement learning (RL) provides a principled and solid framework for sequential decision-making under uncertainty. By incorporating the approximation capacity of deep neural networks, deep reinforcement learning (DRL) has found impressive applications in robotics (Levine et al., 2016), large generative models (OpenAI, 2023), and autonomous driving (Kiran et al., 2021), and obtained super-human performance in tasks such as Go (Silver et al., 2016) and Gran Turismo (Wurman et al., 2022).

However, an RL agent is subject to various types of attacks, including state and reward perturbation, action space manipulation, and model inference and poisoning (Ilahi et al., 2022). Recent studies have shown that an RL agent can be manipulated by poisoning its observation (Huang et al., 2017; Zhang et al., 2020a) and reward signals (Huang & Zhu, 2019), and a well-trained RL agent can be easily defeated by a malicious opponent behaving unexpectedly (Gleave et al., 2020). In particular, recent research has demonstrated the brittleness (Zhang et al., 2020a; Sun et al., 2021) of existing RL algorithms in the face of adversarial state perturbations, where a malicious agent strategically and stealthily perturbs the observations of a trained RL agent, causing a significant loss of cumulative reward. Such an attack can be implemented in practice by exploiting the defects in the agent's perception component, e.g., sensors and communication channels. This raises significant concerns when applying RL techniques in security and safety-critical domains.

Several solutions have been proposed to combat state perturbation attacks. Among them, SA-MDP (Zhang et al., 2020a) imposes a regularization term in the training objective to improve the smoothness of the learned policy under state perturbations. This approach is improved in WocaR-RL (Liang et al., 2022) by incorporating an estimate of the worst-case reward under attacks into the training objective. In a different direction, ATLA (Zhang et al., 2021) alternately trains the agent's policy and the attacker's perturbation policy, utilizing the fact that under a fixed agent policy, the attacker's problem of finding the optimal perturbations can be viewed as a Markov decision process

(MDP) and solved by RL. This approach can potentially lead to a more robust policy but incurs high computational overhead, especially for large environments such as Atari games with raw pixel observations.

Despite their promising performance in certain RL environments, the above solutions have two major limitations. First, actions are directly derived from a value or policy network trained using true states, despite the fact that the agent can only observe perturbed states at the test stage. This mismatch between the training and testing leads to unstable performance at the test stage. Second, most existing work does not exploit historical observations and the agent's knowledge about the underlying MDP model to characterize and reduce uncertainty and infer true states in a systematic way.

In this work, we propose a pessimistic DQN algorithm against state perturbations by viewing the defender's problem as finding an approximate Stackelberg equilibrium for a two-player Markov game with asymmetric observations. Given a perturbed state, the agent selects an action that maximizes the worst-case value across possible true states. This approach is applied at both training and test stages, thus removing the inconsistency between the two. We further propose two approaches to reduce the agent's uncertainty about true states. First, the agent maintains a belief about the actual state using historical data, which, together with the pessimistic approach, provides a strong defense against large perturbations that may change the semantics of states. Second, for games with raw pixel input, such as Atari games, we train a diffusion model using the agent's knowledge about valid states, which is then used to purify observed states. This approach provides superb performance under commonly used attacks, with the additional advantage of being agnostic to the perturbation level. Our method achieves high robustness and significantly outperforms existing solutions under strong attacks while maintaining comparable performance under relatively weak attacks. Further, its training complexity is comparable to SA-MDP and WocaR-RL and is much lower than alternating training-based approaches.

## 2 BACKGROUND

### 2.1 REINFORCEMENT LEARNING

A reinforcement learning environment is usually formulated as a Markov Decision Process (MDP), denoted by a tuple $\langle S, A, P, R, \gamma \rangle$, where $S$ is the state space and $A$ is the action space. $P : S \times A \to \Delta(S)$ is the transition function of the MDP, where $P(s'|s, a)$ gives the probability of moving to state $s'$ given the current state $s$ and action $a$. $R : S \times A \to \mathbb{R}$ is the reward function where $R(s, a) = \mathbb{E}(R_t|s_{t-1} = s, a_{t-1} = a)$ and $R_t$ is the reward in time step $t$. Finally, $\gamma$ is the discount factor. An RL agent wants to maximize its cumulative reward $G = \Sigma_{t=0}^{T} \gamma^t R_t$ over a time horizon $T \in \mathbb{Z}^+ \cup \{\infty\}$, by finding a (stationary) policy $\pi : S \to \Delta(A)$, which can be either deterministic or stochastic. For any policy $\pi$, the state-value and action-value functions are two standard ways to measure how good $\pi$ is. The state-value function satisfies the Bellman equation $V_\pi(s) = \Sigma_{a \in A} \pi(a|s)[R(s, a) + \gamma \Sigma_{s' \in S} P(s'|s, a) V_\pi(s')]$ and the action-value function satisfies $Q_\pi(s, a) = R(s, a) + \gamma \Sigma_{s' \in S} P(s'|s, a)[\Sigma_{a' \in A} \pi(a'|s') Q_\pi(s', a')]$. For MDPs with a finite or countably infinite state space and a finite action space, there is a deterministic and stationary policy that is simultaneously optimal for all initial states $s$. For large and continuous state and action spaces, deep reinforcement learning (DRL) incorporates the powerful approximation capacity of deep learning into RL and has found notable applications in various domains.

### 2.2 STATE ADVERSARIAL ATTACKS IN RL

First introduced in Huang et al. (2017), a state perturbation attack is a test stage attack targeting an agent with a well-trained policy $\pi$. At each time step, the attacker observes the true state $s_t$ and generates a perturbed state $\tilde{s}_t$ (see Figure 1 for examples). The agent observes $\tilde{s}_t$ but not $s_t$ and takes an action $a_t$ according to $\pi(\cdot|\tilde{s}_t)$. The attacker's goal is to minimize the cumulative reward that the agent obtains. Note that the attacker only interferes with the agent's observed state but not the underlying MDP. Thus, the true

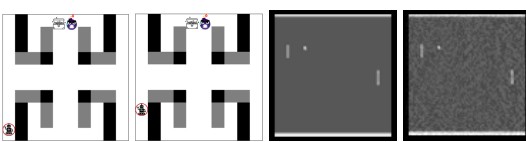

(a) Original (b) Perturbed (c) Original (d) Perturbed

Figure 1: Examples of perturbed states : (a) and (b) show states in a continuous state Gridworld, and (c) and (d) show states in the Atari Pong game.

state in the next time step is distributed according to $P(s_{t+1}|s_t, \pi(\cdot|\tilde{s}_t))$. To limit the attacker's capability and avoid being detected, we assume that $\tilde{s}_t \in B_\epsilon(s_t)$ where $B_\epsilon(s_t)$ is the $l_p$ ball centered at $s_t$ for some norm $p$. We consider a strong adversary that has access to both the MDP and the agent's policy $\pi$ and can perturb at every time step. With these assumptions, it is easy to see that the attacker's problem given a fixed $\pi$ can also be formulated as an MDP $\langle S, S, \tilde{P}, \tilde{R}, \gamma \rangle$, where both the state and action spaces are $S$, the transition probability $\tilde{P}(s'|s, \tilde{s}) = \sum_a \pi(a|\tilde{s})P(s'|s, a)$, and reward $\tilde{R}(s, \tilde{s}) = -\sum_a \pi(a|\tilde{s})R(s, a)$. Thus, an RL algorithm can be used to find a (nearly) optimal attack policy. Further, we adopt the common assumption (Zhang et al., 2020a; Liang et al., 2022) that the agent has access to an intact MDP at the training stage and has access to $\epsilon$ (or an estimation of it). As we discuss below, our diffusion-based approach is agnostic to $\epsilon$. Detailed discussions of related work on attacks and defenses in RL, including and beyond state perturbation, are in Appendix B.

## 3 PESSIMISTIC Q-LEARNING WITH STATE INFERENCE AND PURIFICATION

In this section, we first formulate the robust RL problem as a two-player Stackelberg Markov game. We then present our pessimistic Q-learning algorithm that derives maximin actions from the Q-function using perturbed states as the input to safeguard against the agent's uncertainty about true states. We further incorporate a belief state approximation scheme and a diffusion-based state purification scheme into the algorithm to reduce uncertainty. Our extensions of the vanilla DQN algorithm that incorporates all three mechanisms are given in Algorithms 4-7 in Appendix E. We further give a theoretical result that characterizes the performance loss of being pessimistic.

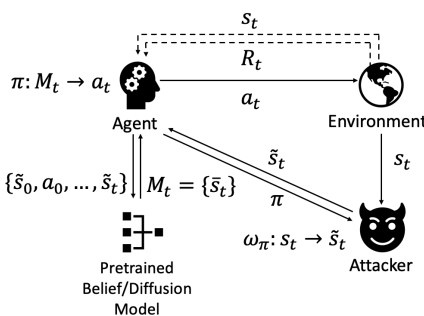

Figure 2: Belief-enriched robust RL against state perturbations. Note that the agent can only access the true state $s_t$ and reward $R_t$ at the training stage.

### 3.1 STATE-ADVERSARIAL MDP AS A STACKELBERG MARKOV GAME WITH ASYMMETRIC OBSERVATIONS

The problem of robust RL under adversarial state perturbations can be viewed as a two-player Markov game, which motivates our pessimistic Q-learning algorithm given in the next subsection. The two players are the RL agent and the attacker with their state and action spaces and reward functions described in Section 2.2. The RL agent wants to find a policy $\pi : S \rightarrow \Delta(A)$ that maximizes its long-term return, while the attacker wants to find an attack policy $\omega : S \rightarrow S$ to minimize the RL agent's cumulative reward. The game has asymmetric observations in that the attacker can observe the true states while the RL agent observes the perturbed states only. The agent's value functions for a given pair of policies $\pi$ and $\omega$ satisfy the Bellman equations below.

**Definition 1.** Bellman equations for state and action value functions under a state adversarial attack:

$$V_{\pi \circ \omega}(s) = \Sigma_{a \in A}\pi(a|(\omega(s))[R(s, a) + \gamma\Sigma_{s' \in S}P(s'|s, a)V_{\pi \circ \omega}(s')];$$
$$Q_{\pi \circ \omega}(s, a) = R(s, a) + \gamma\Sigma_{s' \in S}P(s'|s, a)[\Sigma_{a' \in A}\pi(a'|\omega(s'))Q_{\pi \circ \omega}(s', a')].$$

To achieve robustness, a common approach is to consider a Stackelberg equilibrium by viewing the RL agent as the leader and the attacker as the follower. The agent first commits to a policy $\pi$. The attack observes $\pi$ and identifies an optimal attack, denoted by $\omega_\pi$, as a response, where $\omega_\pi(s) = \operatorname{argmin}_{\tilde{s} \in B_{\epsilon(s)}}\Sigma_{a' \in A}\pi(a'|\tilde{s})Q(s, a')$. As the agent has access to the intact environment at the training stage and the attacker's budget $\epsilon$, it can, in principle, identify a robust policy proactively by simulating the attacker's behavior. Ideally, the agent wants to find a policy $\pi^*$ that reaches a Stackleberg equilibrium of the game, which is defined as follows.

**Definition 2.** A policy $\pi^*$ is a Stackelberg equilibrium of a Markov game if

$$\forall s \in S, \forall \pi, V_{\pi^* \circ \omega_{\pi^*}}(s) \geq V_{\pi \circ \omega_\pi}(s).$$

A Stackelberg equilibrium ensures that the agent's policy $\pi^*$ is optimal (for any initial state) against the strongest possible *adaptive* attack and, therefore, provides a robustness guarantee. However, previous work has shown that due to the noisy observations, finding a stationary policy that is optimal

for every initial state is generally impossible (Zhang et al., 2020a). Existing solutions either introduce a regularization term to improve the smoothness of the policy or alternatively train the agent's policy and attacker's policy. In this paper, we take a different path with the goal of finding an approximate Stackelberg equilibrium (the accurate definition is in Appendix D.3), which is further improved through state prediction and denoising. Figure 2 shows the high-level framework of our approach, which is discussed in detail below.

## 3.2 STRATEGY I - PESSIMISTIC Q-LEARNING AGAINST THE WORST CASE

Both value-based (Könönen, 2004) and policy-based (Zheng et al., 2022; Vu et al., 2022) approaches have been studied to identify the Stackelberg equilibrium (or an approximation of it) of a Markov game. In particular, Stackelberg Q-learning (Könönen, 2004) maintains separate Q-functions for the leader and the follower, which are updated by solving a stage game associated with the true state in each time step. However, these approaches do not apply to our problem as they all require both players to have access to the true state in each time step. In contrast, the RL agent can only observe the perturbed state. Thus, it needs to commit to a policy for all states centered around the observed state instead of a single action, as in the stage game of Stackelberg Q-learning.

In this work, we present a pessimistic Q-learning algorithm (see Algorithm 1) to address the above challenge. The algorithm maintains a Q-function with the true state as the input, similar to vanilla $Q$-learning. But instead of using a greedy approach to derive the target policy or a $\epsilon$-greedy approach to derive the behavior policy from the Q-function, a $\mathrm{maximin}$ approach is used in both cases. In particular, the target policy is defined as follows (line 5). Given a perturbed state $\tilde{s}$, the agent picks an action that maximizes the worst-case $Q$-value across all possible states in $B_\epsilon(\tilde{s})$, which represents the agent's uncertainty. We abuse the notation a bit and let $\pi(\cdot)$ denote a deterministic policy in the rest of the paper since we focus on Q-learning-based algorithms in this paper. The behavioral policy is defined similarly by adding exploration (lines 8 and 9). The attacker's policy $\omega_\pi$ is derived as the best response to the agent's policy (line 6), where a perturbed state is derived by minimizing the $Q$ value given the agent's policy.

A few remarks follow. First, the $\mathrm{maximin}$ scheme is applied when choosing an action with exploration (line 9) and when updating the Q-function (line 11), and a perturbed state is used as the input in both cases. In contrast, in both SA-DQN (Zhang et al., 2020a) and WocaR-DQN (Liang et al., 2022), actions are obtained from the Q-network using true states at the training stage, while the same network is used at the test stage to derive actions from perturbed states. Our approach removes this inconsistency, leading to better performance, especially under relatively large perturbations. Second, instead of the pessimistic approach, we may also consider maximizing the average case or the best case across $B_\epsilon(\tilde{s})$ when deriving actions, which provides a different tradeoff between robustness and efficiency. Third, we show how policies are derived from the Q-function to help explain the idea of the algorithm. Only the Q-function needs to be maintained when implementing the algorithm.

Figure 3 in the Appendix C illustrates the relations between a true state $s$, the perturbed state $\tilde{s}$, the worst-case state $\bar{s} \in B_\epsilon(\tilde{s})$ for which the action is chosen (line 5). In particular, it shows that the true state $s$ must land in the $\epsilon$-ball centered at $\tilde{s}$, and the worst-case state the RL agent envisions is at most $2\epsilon$ away from the true state. This gap causes performance loss that will be studied in Section 3.6. For environments with large state and action spaces, we apply the above idea to derive pessimistic DQN algorithms (see Algorithms 4- 7 in Appendix E), which further incorporate state inference and purification discussed below. Although we focus on value-based approaches in this work, the key ideas can also be incorporated into Stackelberg policy gradient (Vu et al., 2022) and Stackelberg actor-critic (Zheng et al., 2022) approaches, which is left to our future work.

## 3.3 STRATEGY II - REDUCING UNCERTAINTY USING BELIEFS

In Algorithm 1, the agent's uncertainty against the true state is captured by the $\epsilon$-ball around the perturbed state. A similar idea is adopted in previous regularization-based approaches (Zhang et al., 2020a; Liang et al., 2022). For example, SA-MDP (Zhang et al., 2020a) regulates the maximum difference between the top-1 action under the true state $s$ and that under the perturbed state across all possible perturbed states in $B_\epsilon(s)$. However, this approach is overly conservative and ignores the temporal correlation among consecutive states. Intuitively, the agent can utilize the sequence of historical observations and actions $\{(\tilde{s}_\tau, a_\tau)\}_{\tau < t} \cup \{\tilde{s}_t\}$ and the transition dynamics of the underlying

---

**Algorithm 1:** Pessimistic Q-Learning

---
**Result:** Robust Q-function $Q$
1  Initialize $Q(s, a) = 0$ for all $s \in S$, $a \in A$;
2  **for** *epsiode = 1,2,...* **do**
3     Initialize true state $s$
4     **repeat**
5        Update agent's policy: $\forall \tilde{s} \in S, \pi(\tilde{s}) = \operatorname{argmax}_{a \in A} \min_{\bar{s} \in B_\epsilon(\tilde{s})} Q(\bar{s}, a)$;
6        Update attacker's policy: $\forall s \in S, \omega_\pi(s) = \operatorname{argmin}_{\tilde{s} \in B_\epsilon(s)} Q(s, \pi(\tilde{s}))$;
7        Generate perturbed state $\tilde{s} = \omega_\pi(s)$;
8        Choose $a$ from $\tilde{s}$ using $\pi$ with exploration:
9          $a = \pi(\tilde{s})$ with probability $1 - \epsilon'$; otherwise $a$ is a random action;
10       Take action $a$, observe reward $R$ and next true state $s'$;
11       Update Q-function: $Q(s, a) = Q(s, a) + \alpha\big[R(s, a) + \gamma Q(s', \pi(\omega_\pi(s'))) - Q(s, a)\big]$;
12       $s = s'$;
13    **until** *s is terminal*;
14 **end**

---

MDP to reduce its uncertainty of the current true state $s_t$. This is similar to the belief state approach in partially observable MDPs (POMDPs). The key difference is that in a POMDP, the agent's observation $o_t$ in each time step $t$ is derived from a fixed observation function with $o_t = O(s_t, a_t)$. In contrast, the perturbed state $\tilde{s}_t$ is determined by the attacker's policy $\omega$, which is non-stationary at the training stage and is unknown to the agent at the test stage.

To this end, we propose a simple approach to reduce the agent's worst-case uncertainty as follows. Let $M_t \subseteq B_\epsilon(\tilde{s}_t)$ denote the agent's belief about all possible true states at time step $t$. Initially, we let $M_0 = B_\epsilon(\tilde{s}_0)$. At the end of the time step $t$, we update the belief to include all possible next states that is reachable from the current state and action with a non-zero probability. Formally, let $M'_t = \{s' \in S : \exists s \in M_t, P(s'|s, a_t) > 0\}$. After observing the perturbed state $\tilde{s}_{t+1}$, we then update the belief to be the intersection of $M'_t$ and $B_\epsilon(\tilde{s}_{t+1})$, i.e., $M_{t+1} = M'_t \cap B_\epsilon(\tilde{s}_{t+1})$, which gives the agent's belief at time $t + 1$. Figure 3 in the Appendix C demonstrates this process, and the formal belief update algorithm is given in Algorithm 2 in Appendix E. Our pessimistic Q-learning algorithm can easily incorporate the agent's belief. In each time step $t$, instead of using $B_\epsilon(\tilde{s})$ in Algorithm 1 (line 5), the current belief $M_t$ can be used. It is an interesting open problem to develop a strong attacker that can exploit or even manipulate the agent's belief.

**Belief approximation in large state space environments.** When the state space is high-dimensional and continuous, computing the accurate belief as described above becomes infeasible as computing the intersection between high-dimensional spaces is particularly hard. Previous studies have proposed various techniques to approximate the agent's belief about true states using historical data in partially observable settings, including using classical RNN networks (Ma et al., 2020) and flow-based recurrent belief state learning (Chen et al., 2022). In this work, we adapt the particle filter recurrent neural network (PF-RNN) technique developed in (Ma et al., 2020) to our setting due to its simplicity. In contrast to a standard RNN-based belief model $B : (S \times A)^t \to H$ that maps the historical observations and actions to a deterministic latent state $h_t$, PF-RNN approximates the belief $b(h_t)$ by $\kappa_p$ weighted particles in parallel, which are updated using the particle filter algorithm according to the Bayes rule. An output function $f_{out}$ then maps the weighted average of these particles in the latent space to a prediction of the true state in the original state space.

To apply PF-RNN to our problem, we first train the RNN-based belief model $N$ and the prediction function $f_{out}$ before learning a robust RL policy. This is achieved by using $C$ trajectories generated by a random agent policy and a random attack policy in an intact environment. Then at each time step $t$ during the RL training and testing, we use the belief model $N$ and historical observations and actions to generate $\kappa_p$ particles, map each of them to a state prediction using $f_{out}$, and take the set of $\kappa_p$ predicted states as the belief $M_t$ about the true state. PF-RNN includes two versions that support LSTM and GRU, respectively, and we use PF-LSTM to implement our approach. We define the complete belief model utilizing PF-RNN as $N_p \triangleq f_{out} \circ B$.

We remark that previous work has also utilized historical data to improve robustness. For example, Xiong et al. (2023) uses an LSTM-autoencoder to detect and denoise abnormal states at the test stage, and Zhang et al. (2021) considers an LSTM-based policy in alternating training. However, none of them explicitly approximate the agent's belief about true states and use it to derive a robust policy.

### 3.4 STRATEGY III - PURIFYING INVALID OBSERVATIONS VIA DIFFUSION

For environments that use raw pixels as states, such as Atari Games, perturbed states generated by adding bounded noise to each pixel are mostly "invalid" in the following sense. Let $S_0 \subseteq S$ denote the set of possible initial states. Let $S^0$ denote the set of states that are reachable from any initial state in $S_0$ by following an arbitrary policy. Then perturbed states will fall outside of $S^0$ with high probability. This is especially the case for $l_\infty$ attacks that bound the perturbation applied to each pixel as commonly assumed in existing work (see Appendix C.2 for an example). This observation points to a fundamental limitation of existing perturbation attacks that can be utilized by an RL agent to develop a more efficient defense.

One way to exploit the above observation is to identify a set of "valid" states near a perturbed state and use that as the belief of the true state. However, it is often difficult to check if a state is valid or not and to find such a set due to the fact that raw pixel inputs are usually high-dimensional. Instead, we choose to utilize a diffusion model to purify the perturbed states, which obtains promising performance, as we show in our empirical results.

To this end, we first sample $C'$ trajectories from a clean environment using a pre-trained policy without attack to estimate a state distribution $q(\cdot)$, which is then used to train a Denoising Diffusion Probabilistic Model (DDPM) (Ho et al., 2020). Then during both RL training and testing, when the agent receives a perturbed state $\tilde{s}$, it applies the reverse process of the diffusion model for $k$ steps to generate a set of purified states as the belief $M_t$ of size $\kappa_d$, where $k$ and $\kappa_d$ are hyperparameters. We let $N_d : S \to S^{\kappa_d}$ denote a diffusion-based belief model. Note that rather than starting from random noise in the reverse process as in image generation, we start from a perturbed state that the agent receives and manually add a small amount of pixel-wise noise $\phi$ to it before denoising, inspired by denoised smoothing in deep learning (Xiao et al., 2022). We observe in experiments that using a large $k$ does not hurt performance, although it increases the running time. Thus, unlike previous work, this approach is agnostic to the accurate knowledge of attack budget $\epsilon$. One problem with DDPM, however, is that it incurs high overhead to train the diffusion model and sample from it, making it less suitable for real-time decision-making. To this end, we further evaluate a recently developed fast diffusion technique, Progressive Distillation (Salimans & Ho, 2022), which distills a multi-step sampler into a few-step sampler. As we show in the experiments, the two diffusion models provide different tradeoffs between robustness and running time. A more detailed description of the diffusion models and our adaptations are given in Appendix B.6.

### 3.5 PESSIMISTIC DQN WITH APPROXIMATE BELIEFS AND STATE PURIFICATION

Built upon the above ideas, we develop two pessimistic versions of the classic DQN algorithm (Mnih et al., 2013) by incorporating approximate belief update and diffusion-based purification, denoted by BP-DQN and DP-DQN, respectively. The details are provided in Algorithms 4- 7 in Appendix E. Below we highlight the main differences between our algorithms and vanilla DQN.

The biggest difference lies in the loss function, where we incorporate the maximin search into the loss function to target the worst case. Concretely, instead of setting $y_i = R_i + \gamma \max_{a' \in A} Q'(s_i, a')$ as in vanilla DQN, we set $y_i = R_i + \gamma \max_{a' \in A} \min_{m \in M_i} Q'(m, a')$ where $R_i, s_i, M_i$ are sampled from the replay buffer and $Q'$ is the target network. Similarly, instead of generating actions using the $\epsilon$-greedy (during training) or greedy approaches (during testing), the maximin search is adopted.

To simulate the attacker's behavior, one needs to identify the perturbed state $\tilde{s}$ that minimizes the $Q$ value under the current policy $\pi$ subject to the perturbation constraint. As finding the optimal attack under a large state space is infeasible, we solve the attacker's problem using projected gradient descent (PGD) with $\eta$ iterations to find an approximate attack similar to the PGD attack in (Zhang et al., 2020a). In BP-DQN where approximate beliefs are used, the history of states and actions is saved to generate the belief in each round. In DP-DQN where diffusion is used, the reverse process is applied to both perturbed and true states. That is, the algorithm keeps the purified version of the true

states instead of the original states in the replay buffer during training. We find this approach helps reduce the gap between purified states and true states. In both cases, instead of training a robust policy from scratch, we find that it helps to start with a pre-trained model obtained from an attack-free MDP.

We want to highlight that BP-DQN is primarily designed for environments with structural input, whereas DP-DQN is better suited for environments with raw pixel input. Both approaches demonstrate exceptional performance in their respective scenarios, even when faced with strong attacks, as shown in our experiments. Thus, although combining the two methods by integrating history-based belief and diffusion techniques may seem intuitive, this is only needed when confronted with an even more formidable attacker, such as one that alters both semantic and pixel information in Atari games.

## 3.6 Bounding Performance Loss due to Pessimism

In this section, we characterize the impact of being pessimistic in selecting actions. To obtain insights, we choose to work on a pessimistic version of the classic value iteration algorithm (see Algorithm 3 in Appendix E), which is easier to analyze than the Q-learning algorithm presented in Algorithm 1. To this end, we first define the Bellman operator for a given pair of policies.

**Definition 3.** For a given pair of agent policy $\pi$ and attack policy $\omega$, the Bellman operator for the Q-function is defined as follows.

$$T^{\pi \circ \omega} Q(s,a) = R(s,a) + \gamma \Sigma_{s' \in S} P(s'|s,a) Q(s', \pi(\omega(s'))) \tag{1}$$

The algorithm maintains a Q-function, which is initialized to 0 for all state-action pairs. In each round $n$, the algorithm first derives the agent's policy $\pi_n$ and attacker's policy $\omega_{\pi_n}$ from the current Q-function $Q_n$ in the same way as in Algorithm 1, using the worst-case belief, where

$$\pi_n(\tilde{s}) = \text{argmax}_{a \in A} \min_{\bar{s} \in B_\epsilon(\tilde{s})} Q_n(\bar{s}, a), \forall \tilde{s} \in S.$$
$$\omega_{\pi_n}(s) = \text{argmin}_{\tilde{s} \in B_\epsilon(s)} Q_n(s, \pi_n(\tilde{s})), \forall s \in S.$$

That is, $\pi_n$ is obtained by solving a $\text{maximin}$ problem using the current $Q_n$, and $\omega_{\pi_n}$ is a best response to $\pi_n$. The Q-function is then updated as $Q_{n+1} = T^{\pi_n \circ \omega_{\pi_n}} Q_n$. It is important to note that although $T^{\pi \circ \omega_\pi}$ is a contraction for a fixed $\pi$ (see Lemma 3 in Appendix D for a proof), $T^{\pi_n \circ \omega_{\pi_n}}$ is typically not due its dependence on $Q_n$. Thus, $Q_n$ may not converge in general, which is consistent with the known fact that a state-adversarial MDP may not have a stationary policy that is optimal for every initial state. However, we show below that we can still bound the gap between the Q-value obtained by following the joint policy $\tilde{\pi}_n := \pi_n \circ \omega_{\pi_n}$, denoted by $Q^{\tilde{\pi}_n}$, and the optimal Q-value for the original MDP without attacks, denoted by $Q^*$. It is known that $Q^*$ is the unique fixed point of the Bellman optimal operator $T^*$, i.e., $T^* Q^* = Q^*$, where

$$T^* Q(s,a) = R(s,a) + \gamma \Sigma_{s' \in S} P(s'|s,a) \max_{a' \in A} Q(s', a'). \tag{2}$$

We first make the following assumptions about the reward and transition functions of an MDP and then state the main result after that.

**Assumption 1.** The reward function and transition function are Lipschitz continuous. That is, there are constants $l_r$ and $l_p$ such that for $\forall s_1, s_2, s' \in S, \forall a \in A$, we have

$$|R(s_1, a) - R(s_2, a)| \le l_r \|s_1 - s_2\|, |P(s'|s_1, a) - P(s'|s_2, a)| \le l_p \|s_1 - s_2\|.$$

**Assumption 2.** Reward $R$ is upper bounded where for any $s \in S$ and $a \in A$, $R(s,a) \le R_{max}$.

**Theorem 1.** *The gap between $Q^{\tilde{\pi}_n}$ and $Q^*$ is bounded by*

$$\text{limsup}_{n \to \infty} \|Q^* - Q^{\tilde{\pi}_n}\|_\infty \le \frac{1+\gamma}{(1-\gamma)^2} \Delta,$$

*where $\tilde{\pi}_n$ is obtained by Algorithm 3 and $\Delta = 2\epsilon\gamma(l_r + l_p |S| \frac{R_{max}}{1-\gamma})$.*

We give a proof sketch and leave the detailed proof in Appendix D. We first show that $Q^{\tilde{\pi}_n}$ is Lipschitz continuous using Assumption 1. Then we establish a bound of $\|T^* Q_n - Q_{n+1}\|_\infty$ and prove that $T^{\pi \circ \omega_\pi}$ for a fixed policy $\pi$ is a contraction. Finally, we prove Theorem 1 following the idea of Proposition 6.1 in (Bertsekas & Tsitsiklis, 1996).

| Environment | Model | Natural Reward | PGD | | MinBest | | PA-AD | |
|---|---|---|---|---|---|---|---|---|
| | | | $\epsilon = 0.1$ | $\epsilon = 0.5$ | $\epsilon = 0.1$ | $\epsilon = 0.5$ | $\epsilon = 0.1$ | $\epsilon = 0.5$ |
| GridWorld Continous | DQN | $156.5 \pm 90.2$ | $128 \pm 118$ | $-53 \pm 86$ | $98.2 \pm 137$ | $98.2 \pm 137$ | $-10.7 \pm 136$ | $-35.9 \pm 118$ |
| | SA-DQN | $20.8 \pm 140$ | $46 \pm 142$ | $-100 \pm 0$ | $-5.8 \pm 131$ | $-100 \pm 0$ | $-97.5 \pm 13.6$ | $-67.8 \pm 78.3$ |
| | WocaR-DQN | $-63 \pm 88$ | $-100 \pm 0$ | $-63.2 \pm 88$ | $-100 \pm 0$ | $-63.2 \pm 88$ | $-100 \pm 0$ | $-63.2 \pm 88$ |
| | BP-DQN (Ours) | $163 \pm 26$ | $165 \pm 29$ | $176 \pm 16$ | $147 \pm 88$ | $114 \pm 114$ | $171.9 \pm 17$ | $177.2 \pm 10.6$ |

(a) Continuous Gridworld Results

| Env | Model | Natural Reward | PGD | | | MinBest | | | PA-AD | | |
|---|---|---|---|---|---|---|---|---|---|---|---|
| | | | $\epsilon = 1/255$ | $\epsilon = 3/255$ | $\epsilon = 15/255$ | $\epsilon = 1/255$ | $\epsilon = 3/255$ | $\epsilon = 15/255$ | $\epsilon = 1/255$ | $\epsilon = 3/255$ | $\epsilon = 15/255$ |
| Pong | DQN | $21 \pm 0$ | $-21 \pm 0$ | $-21 \pm 0$ | $-21 \pm 0$ | $-21 \pm 0$ | $-21 \pm 0$ | $-21 \pm 0$ | $-18.2 \pm 2.3$ | $-19 \pm 2.2$ | $-21 \pm 0$ |
| | SA-DQN | $21 \pm 0$ | $21 \pm 0$ | $21 \pm 0$ | $-20.8 \pm 0.4$ | $21 \pm 0$ | $21 \pm 0$ | $-21 \pm 0$ | $21 \pm 0$ | $18.7 \pm 2.6$ | $-20 \pm 0$ |
| | WocaR-DQN | $21 \pm 0$ | $21 \pm 0$ | $21 \pm 0$ | $-21 \pm 0$ | $21 \pm 0$ | $21 \pm 0$ | $-21 \pm 0$ | $21 \pm 0$ | $19.7 \pm 2.4$ | $-21 \pm 0$ |
| | DP-DQN-O (Ours) | $19.9 \pm 0.3$ | $19.9 \pm 0.3$ | $19.8 \pm 0.4$ | $19.7 \pm 0.5$ | $19.9 \pm 0.3$ | $19.9 \pm 0.3$ | $19.3 \pm 0.8$ | $19.9 \pm 0.3$ | $19.9 \pm 0.3$ | $19.3 \pm 0.8$ |
| | DP-DQN-F (Ours) | $20.8 \pm 0.4$ | $20.4 \pm 0.9$ | $20.4 \pm 0.9$ | $18.3 \pm 1.9$ | $20.6 \pm 0.9$ | $20.4 \pm 0.8$ | $21.0 \pm 0.0$ | $18.6 \pm 2.5$ | $20.0 \pm 1$ | $18.2 \pm 1.8$ |
| Freeway | DQN | $34 \pm 0.1$ | $0 \pm 0$ | $0 \pm 0$ | $0 \pm 0$ | $0 \pm 0$ | $0 \pm 0$ | $0 \pm 0$ | $0 \pm 0$ | $0 \pm 0$ | $0 \pm 0$ |
| | SA-DQN | $30 \pm 0$ | $30 \pm 0$ | $30 \pm 0$ | $0 \pm 0$ | $27.2 \pm 3.4$ | $18.3 \pm 3.0$ | $0 \pm 0$ | $20.1 \pm 4.0$ | $9.5 \pm 3.8$ | $0 \pm 0$ |
| | WocaR-DQN | $31.2 \pm 0.4$ | $31.2 \pm 0.5$ | $31.4 \pm 0.3$ | $21.6 \pm 1$ | $29.6 \pm 2.5$ | $19.8 \pm 3.8$ | $21.6 \pm 1$ | $24.9 \pm 3.7$ | $12.3 \pm 3.2$ | $21.6 \pm 1$ |
| | DP-DQN-O (Ours) | $28.8 \pm 1.1$ | $29.1 \pm 1.1$ | $29 \pm 0.9$ | $28.9 \pm 0.7$ | $29.2 \pm 1.0$ | $28.5 \pm 1.2$ | $28.6 \pm 1.3$ | $28.6 \pm 1.2$ | $28.3 \pm 1$ | $28.8 \pm 1.3$ |
| | DP-DQN-F (Ours) | $31.2 \pm 1$ | $30.0 \pm 0.9$ | $30.1 \pm 1$ | $30.7 \pm 1.2$ | $30.2 \pm 1.3$ | $30.6 \pm 1.4$ | $29.4 \pm 1.2$ | $30.8 \pm 1$ | $31.4 \pm 0.8$ | $28.9 \pm 1.1$ |

(b) Atari Games Results

Table 1: Experiment Results. We show the average episode rewards $\pm$ standard deviation over 10 episodes for our methods and three baselines. The results for our methods are highlighted in gray.

## 4 EXPERIMENTS

In this section, we evaluate our belief-enriched pessimistic DQN algorithms by conducting experiments on three environments, a continuous state Gridworld environment (shown in Figure 1a) for BP-DQN and two Atari games, Pong and Freeway for DP-DQN-O and DP-DQN-F, which utilize DDPM and Progressive Distillation as the diffusion model, respectively. (See Appendix F.1 for a justification.) We choose vanilla DQN (Mnih et al., 2015), SA-DQN (Zhang et al., 2020a), WocaR-DQN (Liang et al., 2022), and Radial-DQN (Oikarinen et al., 2021) as defense baselines. We consider three commonly used attacks to evaluate the robustness of these algorithms: (1) PGD attack (Zhang et al., 2020a), which aims to find a perturbed state $\tilde{s}$ that minimizes $Q(s, \pi(\tilde{s}))$; (2) MinBest attack (Huang et al., 2017), which aims to find a perturbed state $\tilde{s}$ that minimizes the probability of choosing the best action under $s$; and (3) PA-AD (Sun et al., 2021), which utilizes RL to find a (nearly) optimal attack policy. Details on the environments and experiment setup can be found in Appendix F.2. Additional experiment results and ablation studies are given in Appendix F.3.

### 4.1 RESULTS AND DISCUSSION

**Continuous Gridworld.** As shown in Table 1a, our method (BP-DQN) achieves the best performance under all scenarios in the continuous state Gridworld environment and significantly surpasses all the baselines. In contrast, both SA-DQN and WocaR-DQN fail under the large attack budget $\epsilon = 0.5$ and perform poorly under the small attack budget $\epsilon = 0.1$. We conjecture that this is because state perturbations in the continuous Gridworld environment often change the semantics of states since most perturbed states are still valid observations. We also noticed that both SA-DQN and WocaR-DQN perform worse than vanilla DQN when there is no attack and when $\epsilon = 0.1$. We conjecture that this is due to the mismatch between true states and perturbed states during training and testing and the approximation used to estimate the upper and lower bounds of Q-network output using the Interval Bound Propagation (IBP) technique (Gowal et al., 2019) in their implementations. Although WocaR-DQN performs better under $\epsilon = 0.5$ than $\epsilon = 0.1$, it fails in both cases to achieve the goal of the agent where the policies end up wandering in the environment or reaching the bomb instead of finding the gold.

**Atari Games.** As shown in Table 1b, our DP-DQN method outperforms all other baselines under a strong attack (e.g., PA-AD) or a large attack budget (e.g., $\epsilon = 15/255$), while achieving comparable performance as other baselines in other cases. SA-DQN and WocaR-DQN fail to respond to large state perturbations for two reasons. First, both of them use IBP to estimate an upper and lower bound of the neural network output under perturbations, which are likely to be loose under large perturbations. Second, both approaches utilize a regularization-based approach to maximize the chance of choosing the best action for all states in the $\epsilon$-ball centered at the true state. This approach is effective under small perturbations but can pick poor actions for large perturbations as the latter can easily exceed the generalization capability of the Q-network. We observe that WocaR-DQN performs better when the attack budget increases from $3/255$ to $15/255$ in Freeway, which is counter-intuitive.

| Environment | Model | PGD $\epsilon = 0.5$ | Environment | Model | PA-AD $\epsilon = 15/255$ |
|---|---|---|---|---|---|
| **Continuous Gridworld** | Maximin Only | $-71 \pm 91$ | **Pong** | Maximin Only | $-21 \pm 0$ |
| | Belief Only | $45.7 \pm 134$ | | DDPM Only | $18.8 \pm 1.6$ |
| | BP-DQN (Ours) | $176 \pm 16$ | | DP-DQN-O (Ours) | $19.3 \pm 0.8$ |

Table 2: Ablation Study Results. We compare our methods with variants that use $\mathrm{maximin}$ search or belief approximation only.

The reason is that under large perturbations, the agent adopts a bad policy by always moving forward regardless of state, which gives a reward of around 21. Further, SA-DQN and WocaR-DQN need to know the attack budget in advance in order to train their policies. A policy trained under attack budget $\epsilon = 1/255$ is ineffective against larger attack budgets. In contrast, our DP-DQN method is agnostic to the perturbation level. All results of DP-DQN shown in Table 1b are generated with the same policy trained under attack budget $\epsilon = 1/255$.

We admit that our method suffers a small performance loss compared with SA-DQN and WocaR-DQN in the Atari games when there is no attack or when the attack budget is low. We conjecture that no single fixed policy is simultaneously optimal against different types of attacks. A promising direction is to adapt a pre-trained policy to the actual attack using samples collected online.

**Importance of Combining Maximin and Belief.** In Table 2, we compare our methods (BP-DQN and DP-DQN-O) that integrate the ideas of $\mathrm{maximin}$ search and belief approximation (using either RNN or diffusion) with variants of our methods that use $\mathrm{maximin}$ search or belief approximation only. The former is implemented using a trained BP-DQN or DP-DQN-O policy together with random samples from the $\epsilon$-ball centered at a perturbed state (the worst-case belief) during the test stage. The latter uses the vanilla DQN policy with a single belief state generated by either the PF-RNN or the DDPM diffusion model at the test stage. The results clearly demonstrate the importance of integrating both ideas to achieve more robust defenses.

## 5 CONCLUSION AND LIMITATIONS

In conclusion, this work proposes two algorithms, BP-DQN and DP-DQN, to combat state perturbations against reinforcement learning. Our methods achieve high robustness and significantly outperform state-of-the-art baselines under strong attacks. Further, our DP-DQN method has revealed an important limitation of existing state adversarial attacks on RL agents with raw pixel input, pointing to a promising direction for future research.

However, our work also has some limitations. First, our method needs access to a clean environment during training. Although the same assumption has been made in most previous work in this area, including SA-MDP and WocaR-MDP, a promising direction is to consider an offline setting to release the need to access a clean environment by learning directly from (possibly poisoned) trajectory data. Second, using a diffusion model increases the computational complexity of our method and causes slow running speed at the test stage. Fortunately, we have shown that fast diffusion methods can significantly speed up runtime performance. Third, we have focused on value-based methods in this work. Extending our approach to policy-based methods is an important next step.

## ACKNOWLEDGMENTS

This work has been funded in part by NSF grant CNS-2146548 and Tulane University Jurist Center for Artificial Intelligence. We thank the anonymous reviewers for their valuable and insightful feedback.

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

APPENDIX

## A  BROADER IMPACTS

As RL is increasingly being used in vital real-world applications like autonomous driving and large generative models, we are rapidly moving towards an AI-assisted society. With AI becoming more widespread, it is important to ensure that the policies governing AI are robust. An unstable policy could be easily exploited by malicious individuals or organizations, causing damage to property, productivity, and even loss of life. Therefore, providing robustness is crucial to the successful deployment of RL and other deep learning algorithms in the real world. Our work provides new insights into enhancing the robustness of RL policies against adversarial attacks and contributes to the foundation of trustworthy AI.

## B  RELATED WORK

### B.1  STATE PERTURBATION ATTACKS AND DEFENSES

State perturbation attacks against RL policies are first introduced in Huang et al. (2017), where the MinBest attack that minimizes the probability of choosing the best action is proposed. Zhang et al. (2020a) show that when the agent's policy is fixed, the problem of finding the optimal adversarial policy is also an MDP, which can be solved using RL. This approach is further improved in (Sun et al., 2021), where a more efficient algorithm for finding the optimal attack called PA-AD is developed.

On the defense side, Zhang et al. (2020a) prove that a policy that is optimal for any initial state under optimal state perturbation might not exist and propose a set of regularization-based algorithms (SA-DQN, SA-PPO, SA-DDPG) to train a robust agent against state perturbations. This approach is improved in (Liang et al., 2022) by training an additional worst-case Q-network and introducing state importance weights into regularization. In a different direction, an alternating training framework called ATLA is studied in (Zhang et al., 2021) that trains the RL attacker and RL agent alternatively in order to increase the robustness of the DRL model. However, this approach suffers from high computational overhead. Xiong et al. (2023) propose an auto-encoder-based detection and denoising framework to detect perturbed states and restore true states. Also, He et al. (2023) show that when the initial distribution is known, a policy that optimizes the expected return across initial states under state perturbations exists.

### B.2  ATTACKS AND DEFENSES BEYOND STATE PERTURBATIONS

This section briefly introduces other types of adversarial attacks in RL beyond state perturbation. As shown in (Huang & Zhu, 2019), manipulating the reward signal can successfully affect the training convergence of Q-learning and mislead the trained agent to follow a policy that the attacker aims at. Furthermore, an adaptive reward poisoning method is proposed by (Zhang et al., 2020b) to achieve a nefarious policy in steps polynomial in state-space size $|S|$ in the tabular setting.

Lee et al. (2020b) propose two methods for perturbing the action space, where the $LAS$ (look-ahead action space) method achieves better attack performance in terms of decreasing the cumulative reward of DRL by distributing attacks across the action and temporal dimensions. Another line of work investigates adversarial policies in a multi-agent environment, where it has been shown that an opponent adopting an adversarial policy could easily beat an agent with a well-trained policy in a zero-sum game (Gleave et al., 2020).

For attacking an RL agent's policy network, both inference attacks (Chen et al., 2021), where the attacker aims to steal the policy network parameters, and poisoning attacks (Huai et al., 2020) that directly manipulate model parameters have been considered. In particular, an optimization-based technique for identifying an optimal strategy for poisoning the policy network is proposed in (Huai et al., 2020).

### B.3 BACKDOOR ATTACKS IN RL

Recent work investigating defenses against backdoor attacks in RL also considers recovering true states to gain robustness (Bharti et al., 2022). However, there are important differences between our work and (Bharti et al., 2022). First, our work contains two important parts that (Bharti et al., 2022) does not have, which are the maximin formulation and belief update. The former allows us to obtain a robust policy by making fewer assumptions about attack behavior compared to (Bharti et al., 2022). Note that this approach is unique to state-perturbation attacks, as it is difficult to define a worst-case scenario for backdoor attacks. The latter is crucial to combat adaptive perturbations that can change the semantic meaning of states, which can potentially be very useful to backdoor attacks as well. Second, our Lipschitz assumptions differ from those in (Bharti et al., 2022). We assume that the reward and transition functions of the underlying MDP are Lipschitz continuous while Bharti et al. (2022) assume that the backdoored policies are Lipschitz continuous.

### B.4 PARTIALLY OBSERVABLE MDPs

As first proposed by Astrom et al. (1965), a Partially Observable MDP is a generalization of an MDP where the system dynamics are determined by an MDP, but the agent does not have full access to the state. The agent could only partially observe the underlying state that is usually determined by a fixed observation function $\mathcal{O}$. POMDPs could model a lot of real-life sequential decision-making problems such as robot navigation. However, since the agent does not have perfect information about the state, solutions for POMDPs usually need to infer a belief about the true state and find an action that is optimal for each possible belief. To this end, algorithms for finding a compressed belief space in order to solve large state space POMDPs have been proposed (Roy et al., 2005). State-of-the-art solutions approximate the belief states with distributions such as diagonal Gaussian (Lee et al., 2020a), Gaussian mixture (Tschiatschek et al., 2018), categorical distribution (Hafner et al., 2021) or particle filters (Ma et al., 2020). Most recently, a flow-based recurrent belief state modeling approach has been proposed in (Chen et al., 2022) to approximate general continuous belief states.

The main difference between POMDPs and MDPs under state adversarial attacks is the way the agent's observation is determined. In a POMDP, the agent's partial observation at time step $t$ is determined by a fixed observation function $\mathcal{O}$, where $o_t = \mathcal{O}(s_t, a_t)$. And it is independent of the agent's policy $\pi$. Instead, in an MDP under state adversarial attacks, a perturbed state $\tilde{s}$ is determined by the attack policy $\omega$, which can adapt to the agent's policy $\pi$ in general.

### B.5 RL FOR STACKELBERG MARKOV GAMES

Previous work has studied various techniques for solving the Stackelberg equilibrium of asymmetric Markov games, with one player as the leader and the rest being followers. Kononen (Könönen, 2004) proposes an asymmetric multi-agent Q-Learning algorithm and establishes its convergence in the tabular setting. Besides value-based approaches, Fiez et al. (Fiez et al., 2020) recently investigated sufficient conditions for a local Stackelberg equilibrium (LSE) and derived gradient-based learning dynamics for Stackelberg games using the implicit function theorem. Follow-up work applied this idea to derive Stackelberg actor-critic (Zheng et al., 2022) and Stackelberg policy gradient (Vu et al., 2022) methods. However, all these studies assume that the true state information is accessible to all players, which does not apply to our problem.

### B.6 MORE DETAILS ABOUT DIFFUSION-BASED DENOISING

In a DDPM model, the forward process constructs a discrete-time Markov chain as follows. Given an initial state $\mathbf{x}_0$ sampled from $q(\cdot)$, it gradually adds Gaussian noise to $\mathbf{x}_0$ to generate a sequence of noisy states $\mathbf{x}_1, \mathbf{x}_2, ..., \mathbf{x}_K$ where $q(\mathbf{x}_i \mid \mathbf{x}_{i-1}) := \mathcal{N}(\mathbf{x}_i; \sqrt{1 - \beta_i} \mathbf{x}_{i-1}, \beta_i \mathbf{I})$ so that $\mathbf{x}_K$ approximates the Gaussian white noise. Here $\beta_i$ is precalculated according to a variance schedule and $\mathbf{I}$ is the identity matrix. The reverse process is again a Markov chain that starts with $\mathbf{x}_K$ sampled from the Gaussian white noise $\mathcal{N}(0, \mathbf{I})$ and learns to remove the noise added in the forward process to regenerate $q(\cdot)$. This is achieved through the reverse transition $p_\theta(\mathbf{x}_{i-1} \mid \mathbf{x}_i) := \mathcal{N}(\mathbf{x}_{i-1}; \boldsymbol{\mu}_\theta(\mathbf{x}_i, i), \boldsymbol{\Sigma}_\theta(\mathbf{x}_i, i))$ where $\theta$ denotes the network parameters used to approximate the mean and the variance added in the forward process. As mentioned in the main text, we modify the reverse process by starting from a perturbed state $\tilde{s} + \phi$ instead of $\mathbf{x}_K$, where $\phi$ is pixel-wise noise sampled from Gaussian

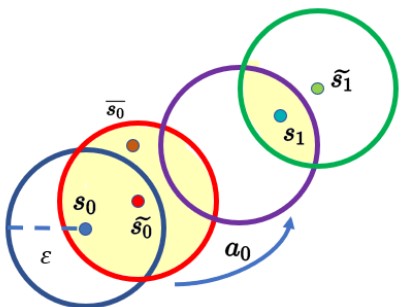

Figure 3: True, perturbed, and worst-case states in Algorithm 1 and belief update. Beginning with true state $s_0$ and perturbed state $\tilde{s}_0$, the agent will have an initial belief, i.e., the $\epsilon$ ball centered at $\tilde{s}_0$. After taking action $a_0$, the belief is updated to the region marked by the purple ball. When observing the next perturbed state $\tilde{s}_1$, the agent will update belief by taking the intersection of the purple ball and the green ball.

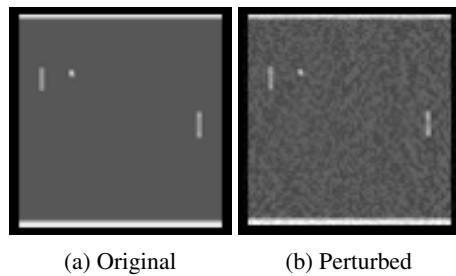

(a) Original          (b) Perturbed

Figure 4: An example of valid vs. invalid states in Pong.

distribution $\mathcal{N}(0, \epsilon_\phi^2)$. We then take $k$ reverse steps with $k \ll K$, according to the observation that a perturbed state only introduces a small amount of noise to the true state due to the attack budget $\epsilon$. We observe in our experiments that using a large $k$ does not hurt the performance, although it increases the running time (see Figures 5d and 5e in Appendix F.3).

The progressive distillation diffusion model (Salimans & Ho, 2022) can distill an $N$ steps sampler to a new sampler of $N/2$ steps with little degradation of sample quality. Thus with a $1024$ step sampler, we could generate $512$ step, $256$ step, ..., and $8$ step samplers. Notice that a single reverse step in an $8$ step sampler will have an equivalent effect of sampling multiple steps in the original $1024$ step sampler. By choosing a proper sampler generated by progressive distillation (we report every model we used in F.2), we could accelerate our diffusion process while preserving sample quality at the same time.

## C   MORE GRAPHS AND EXAMPLES

### C.1   AN EXAMPLE OF BELIEF UPDATE

Figure 3 illustrates the relations between a true state $s$, the perturbed state $\tilde{s}$, and the worst-case state $\bar{s} \in B_\epsilon(\tilde{s})$ for which the action is chosen.

### C.2   AN EXAMPLE OF INVALID STATES IN PIXEL-WISE PERTURBATIONS

For example, the white bar shown in Figure 4 in the Atari Pong game will not change during gameplay and has a grayscale value of $236/255$. However, a pixel-wise state perturbation attack such as PGD with attack budget $\epsilon = 15/255$ will change the pixel values in the white bar to a range of $221/255 - 251/255$ so that the perturbed states become invalid.

# D PROOFS

## D.1 PROOF OF THEOREM 1

In this section, we prove Theorem 1. Recall that $\tilde{\pi} := \pi \circ \omega_\pi$. We first establish the Lipchitz continuity of $Q^{\tilde{\pi}}$, the Q-value when the agent follows policy $\pi$ and the attack follows policy $\omega_\pi$.

**Lemma 1.** $Q^{\tilde{\pi}}$ *is Lipchitz continuous for any* $\pi$ *, i.e.,* $\forall s, s' \in S, \forall a \in A$,

$$|Q^{\tilde{\pi}}(s,a) - Q^{\tilde{\pi}}(s',a)| \leq \mathcal{L}_{\mathcal{Q}_f} \|s - s'\| \tag{3}$$

*where* $L_{Q_f} = l_r + \frac{R_{max}}{1-\gamma}|S|l_p$

*Proof.* Based on the definition of the action value function under state perturbation, we have

$$Q^{\tilde{\pi}}(s,a) = R(s,a) + \Sigma_{s' \in S}P(s'|s,a)\gamma V_{\pi \circ \omega_\pi}(s')$$
$$\leq R(s,a) + \Sigma_{s' \in S}P(s'|s,a)\gamma V_\pi(s')$$

Thus,

$$|Q^{\tilde{\pi}}(s_1,a) - Q^{\tilde{\pi}}(s_2,a)|$$
$$=|R(s_1,a) - R(s_2,a) + \Sigma_{s' \in S}[(P(s'|s_1,a) - P(s'|s_2,a))V_{\pi \circ \omega_\pi}(s')]|$$
$$\leq|R(s_1,a) - R(s_2,a)| + \Sigma_{s' \in S}|[(P(s'|s_1,a) - P(s'|s_2,a))V_\pi(s')]|$$
$$\overset{(a)}{\leq} l_r\|s_1 - s_2\| + \max_{s' \in S}V_\pi(s')|S|P(s'|s_1,a) - P(s'|s_2,a)|$$
$$\overset{(b)}{\leq} l_r\|s_1 - s_2\| + \frac{R_{max}}{1 - \gamma}|S|l_p\|s_1 - s_2\|$$
$$\leq(l_r + \frac{R_{max}}{1 - \gamma}|S|l_p)\|s_1 - s_2\|$$
$$=\mathcal{L}_{\mathcal{Q}_f}\|s_1 - s_2\|$$

where (a) follows from Assumption 1 and (b) follows from Assumptions 1 and 2 and the definition of $V_\pi$. □

**Lemma 2.** *For* $Q_n$ *defined in Algorithm 3, the Bellman approximation error is bounded by*

$$\|T^*Q_n - Q_{n+1}\|_\infty \leq 2\epsilon\gamma(l_r + l_p|S|\frac{R_{max}}{1 - \gamma}) \tag{4}$$

*Proof.*

$$\|T^*Q_n - Q_{n+1}\|_\infty$$
$$= \max_{s \in S, a \in A}|R(s,a) + \gamma\Sigma_{s'}P(s'|s,a)\max_{a \in A}Q_n(s',a) - [R(s,a) + \gamma\Sigma_{s'}P(s'|s,a)Q_n(s', \pi_n(\omega_{\pi_n}(s')))]|$$
$$= \gamma \max_{s \in S, a \in A}\Sigma_{s' \in S}P(s'|s,a)|\max_{a \in A}Q_n(s',a) - Q_n(s', \tilde{\pi}_n(s'))|$$
$$\leq \gamma\max_{s' \in S}|\max_{a \in A}Q_n(s',a) - Q_n(s', \tilde{\pi}_n(s'))|$$

Let $\tilde{s'} = \omega_{\pi_n}(s')$ denote the perturbation of the true state $s'$, and $\tilde{a}$ and $\bar{s'}$ denote the agent's action when observing $\tilde{s'}$ and the worst-case state in $B_\epsilon(\tilde{s'})$ that solves the maximin problem, respectively. We then have

$$Q_n(s', \tilde{a}) \geq Q_n(\bar{s'}, \tilde{a}) = \max_{a \in A}\min_{s \in B_\epsilon(\tilde{s'})} Q_n(s,a). \tag{5}$$

where the first inequality is due to the fact that $\bar{s}'$ obtains the worst-case Q-value under action $\tilde{a}$, across all states in $B_\epsilon(\bar{s}')$ including $s'$. It follows that

$$
\begin{aligned}
\|T^*Q_n - Q_{n+1}\|_\infty &\leq \gamma \max_{s' \in S} |\max_{a \in A} Q_n(s',a) - Q_n(s', \tilde{\pi}(s'))| \\
&\leq \gamma \max_{s' \in S} |\max_{a \in A} Q_n(s',a) - \max_{a \in A} \min_{s \in B_\epsilon(\bar{s}')} Q_n(s,a)| \\
&\leq \gamma \max_{s' \in S, a \in A} |Q_n(s',a) - \min_{s \in B_\epsilon(\bar{s}')} Q_n(s,a)| \\
&\overset{(a)}{\leq} \gamma \max_{s' \in S, a \in A} |Q_n(s',a) - \min_{s \in B_{2\epsilon}(s')} Q_n(s,a)| \\
&\overset{(b)}{\leq} 2\gamma\epsilon\mathcal{L}_{Q_f}
\end{aligned}
$$

where (a) is due to $\|\bar{s}' - s'\| \leq \epsilon$ and (b) follows from Lemma 1. $\qquad\square$

**Lemma 3.** *Given any policy $\tilde{\pi} = \pi \circ \omega_\pi$ where $\pi$ is a fixed policy, $T^{\tilde{\pi}}$ is a contraction.*

*Proof.*

$$
\begin{aligned}
\|T^{\tilde{\pi}}Q_1 - T^{\tilde{\pi}}Q_2\|_\infty &= \max_{s \in S, a \in A} \Sigma_{s' \in S} \gamma P(s'|s,a)|Q_1(s', \pi(\omega(s'))) - Q_2(s', \pi(\omega(s')))| \\
&\leq \max_{s' \in S} \gamma |Q_1(s', \pi(\omega(s'))) - Q_2(s', \pi(\omega(s')))| \\
&\leq \max_{s' \in S, a \in A} \gamma |Q_1(s', a) - Q_2(s', a)| \\
&= \gamma \|Q_1 - Q_2\|_\infty
\end{aligned}
$$

Thus, for any given policy $\pi$, $T^{\tilde{\pi}}$ is a contraction. $\qquad\square$

With Lemmas 2 and 3, we are ready to prove Theorem 1.

**Theorem 1.** *The gap between $Q^{\tilde{\pi}_n}$ and $Q^*$ is bounded by*

$$
\text{limsup}_{n \to \infty} \|Q^* - Q^{\tilde{\pi}_n}\|_\infty \leq \frac{1+\gamma}{(1-\gamma)^2}\Delta
$$

*where $\tilde{\pi}_n$ is obtained by Algorithm 3 and $\Delta = 2\epsilon\gamma(l_r + l_p|S|\frac{R_{max}}{1-\gamma})$.*

*Proof.*

$$
\begin{aligned}
\|Q^* - Q^{\tilde{\pi}_n}\|_\infty &\overset{(a)}{\leq} \|T^*Q^* - T^*Q_n\|_\infty + \|T^*Q_n - T^{\tilde{\pi}_n}Q^{\tilde{\pi}_n}\|_\infty \\
&\leq \|T^*Q^* - T^*Q_n\|_\infty + \|T^*Q_n - T^{\tilde{\pi}_n}Q_n\|_\infty + \|T^{\tilde{\pi}_n}Q_n - T^{\tilde{\pi}_n}Q^{\tilde{\pi}_n}\|_\infty \\
&\overset{(b)}{\leq} \gamma\|Q^* - Q_n\|_\infty + \|T^*Q_n - Q_{n+1}\|_\infty + \gamma(\|Q_n - Q^*\|_\infty + \|Q^* - Q^{\tilde{\pi}_n}\|_\infty)
\end{aligned}
$$

where (a) follows from $Q^*$ is the fixed point of $T^*$, $Q^{\tilde{\pi}_n}$ is the fixed point of $T^{\tilde{\pi}_n}$, and the triangle inequality, and (b) follows from both $T^*$ and $T^{\tilde{\pi}_n}$ (for a fixed $\pi_n$) are contractions. This together with Lemma 2 implies that

$$
\|Q^* - Q^{\tilde{\pi}_n}\|_\infty \leq \frac{2\gamma\|Q^* - Q_n\|_\infty) + \Delta}{1-\gamma} \tag{6}
$$

We then bound $\|Q^* - Q_n\|_\infty$ as follows

$$
\|Q^* - Q_{n+1}\|_\infty \leq \|T^*Q^* - T^*Q_n\|_\infty + \|T^*Q_n - Q_{n+1}\|_\infty \leq \gamma\|Q^* - Q_n\|_\infty + \Delta,
$$

which implies that

$$
\text{limsup}_{n \to \infty} \|Q^* - Q_n\|_\infty \leq \frac{\Delta}{1-\gamma} \tag{7}
$$

Plug this back in equation 6, we have

$$
\text{limsup}_{n \to \infty} \|Q^* - Q^{\tilde{\pi}_n}\|_\infty \leq \frac{1+\gamma}{(1-\gamma)^2}\Delta \tag{8}
$$

where $\Delta = 2\epsilon\gamma(l_r + l_p|S|\frac{R_{max}}{1-\gamma})$. $\qquad\square$

| $Q_1$ | $s_1$ | $s_2$ | $s_3$ | $Q_2$ | $s_1$ | $s_2$ | $s_3$ |
|-------|-------|-------|-------|-------|-------|-------|-------|
| $a_1$ | 12 | 11 | 3 | $a_1$ | 4 | 2 | -2 |
| $a_2$ | 12 | 10 | 2 | $a_2$ | 4 | 0 | -1 |
| $a_3$ | 12 | 8 | 1 | $a_3$ | 4 | 1 | -3 |

Table 3: A counterexample to $T^{\pi_n \circ \omega_{\pi_n}}$ being a contraction

## D.2 A COUNTEREXAMPLE TO $T^{\pi_n \circ \omega_{\pi_n}}$ BEING A CONTRACTION WHEN $\pi_n \circ \omega_{\pi_n}$ IS NOT FIXED

Consider an MDP $\langle S, A, P, R, \gamma \rangle$ where $S = \{s_1, s_2, s_3\}$ and $A = \{a_1, a_2, a_3\}$. Suppose that for any $s \in S$ and $a \in A$, $P(s_2|s, a) = 1$ and $P(s'|s, a) = 0$ for $s' \neq s_2$, and $B_\epsilon(s) = \{s_1, s_2, s_3\}$. Consider the two $Q$-functions shown in Table 3. We have $||Q_1 - Q_2||_\infty = |Q_1(s_2, a_2) - Q_2(s_2, a_2)| = 10$. However, when $\tilde{\pi}_1 = \pi_1 \circ \omega_{\pi_1}$ is derived from $Q_1$ and $\tilde{\pi}_2 = \pi_2 \circ \omega_{\pi_2}$ is derived from $Q_2$,

$$
\begin{aligned}
||T^{\tilde{\pi}_1} Q_1 - T^{\tilde{\pi}_2} Q_2||_\infty &= \max_{s \in S, a \in A} \sum_{s' \in S} \gamma P(s'|s, a) |Q_1(s', \pi_1(\omega_{\pi_1}(s'))) - Q_2(s', \pi_2(\omega_{\pi_2}(s')))| \\
&= \max_{s \in S, a \in A} \gamma |Q_1(s_2, \pi_1(\omega_{\pi_1}(s'))) - Q_2(s, \pi_2(\omega_{\pi_2}(s')))| \\
&\geq \gamma |Q_1(s_2, \pi_1(\omega_{\pi_1}(s_2))) - Q_2(s_2, \pi_2(\omega_{\pi_2}(s_2)))| \\
&\stackrel{(a)}{=} \gamma |Q_1(s_2, a_1) - Q_2(s_2, a_2)| \\
&= \gamma \times 11 \\
&\stackrel{(b)}{>} 10 = ||Q_1 - Q_2||_\infty
\end{aligned}
$$

where (a) is due to the fact that no matter what $\omega_\pi(s_2)$ is, $B_\epsilon(\omega_\pi(s_2)) = \{s_1, s_2, s_3\} = S$, which implies that $\pi_1(\omega_{\pi_1}(s_2)) = \operatorname{argmax}_{a \in A} \min_{\bar{s} \in S} Q_1(\bar{s}, a) = a_1$, and similarly, $\pi_2(\omega_{\pi_2}(s_2)) = a_2$; (b) holds when $\gamma > \frac{10}{11}$. Therefore, $T^{\pi_1 \circ \omega_{\pi_1}}$ is not a contraction.

## D.3 APPROXIMATE STACKELBERG EQUILIBRIUM

As shown in Theorem 4.11 of (He et al., 2023), when the initial state distribution $\Pr(s_0)$ is known, there is an agent policy $\pi^*$ that maximizes the worst-case expected state value against the optimal state perturbation attack, that is

$$
\forall \pi, \Sigma_{s_0 \in S} \Pr(s_0) V_{\pi^* \circ \omega_{\pi^*}}(s_0) \geq \Sigma_{s_0 \in S} \Pr(s_0) V_{\pi \circ \omega_\pi}(s_0) \tag{9}
$$

In particular, when the initial state $s_0$ is fixed, an optimal policy that maximizes the worst-case state value (against the optimal state perturbation attack) exists. Let $V^*(s_0)$ denote this optimal state value. Note that $V^*(s_0)$ is obtained using a different policy for different $s_0$. Then a reasonable definition of an approximate Stackelberg Equilibrium is a policy $\pi$ where its state value under each state $s_0$ is close to $V^*(s_0)$, that is, $|V_{\pi \circ \omega_\pi}(s_0) - V^*(s_0)| \leq \Delta$ for some constant $\Delta$. Note that this condition should hold for each state $s_0$. Also note that we compare $\pi$ with a different optimal policy with respect to each $s_0$, thus bypassing the impossibility result stated in Section 3.1.

In the paper, we further approximated $V^*(s_0)$ by the optimal value without attacks and derived $\Delta$ in this simplified setting (Theorem 1). However, we believe that the general definition given above presents a novel extension of approximate Stackelberg Equilibrium in the standard sense and it is an interesting open problem to derive a policy that is nearly optimal under this new definition.

# E  ALGORITHMS

---

**Algorithm 2:** Belief Update

---

**Data:** Old Belief $M_t$, action $a$, perturbed state $\tilde{s}_{t+1}$.
**Result:** Updated Belief $M_{t+1}$

1   Initialize $M_t'$ to be an empty set
2   **for** $s$ *in* $M_t$ **do**
3     **for** $s'$ *in* $S$ **do**
4       **if** $P(s'|s,a) \neq 0$ **then**
5         Add $s'$ to $M_t'$
6       **end**
7     **end**
8   **end**
9   $M_{t+1} = M_t' \cap B_\epsilon(\tilde{s}_{t+1})$
10   **RETURN** $M_{t+1}$

---

**Algorithm 3:** Pessimistic Q-Iteration

---

**Result:** Robust Q-function $Q$

1   Initialize $Q_0(s,a) = 0$ for all $s \in S, a \in A$;
2   **for** $n = 0, 1, 2, ...$ **do**
3     Update RL agent policy: $\forall \tilde{s} \in S, \pi_n(\tilde{s}) = \operatorname{argmax}_{a \in A} \min_{\bar{s} \in B_\epsilon(\tilde{s})} Q_n(\bar{s}, a)$;
4     Update attacker policy: $\forall s \in S, \omega_{\pi_n}(s) = \operatorname{argmin}_{\tilde{s} \in B_{\epsilon(s)}} Q_n(s, \pi_n(\tilde{s}))$;
5     **for** $s \in S$ **do**
6       **for** $a \in A$ **do**
7         $Q_{n+1}(s,a) = R(s,a) + \gamma \Sigma_{s' \in S} P(s'|s,a) Q_n(s', \pi(\omega_\pi(s')))$;
8       **end**
9     **end**
10   **end**

---

**Algorithm 4:** Belief-Enriched Pessimistic DQN (BP-DQN) Training. We highlight the difference between our algorithm and the vanilla DQN algorithm in brown.

---

**Data:** Number of iterations $T$, trained vanilla Q network $Q_v$, PF-RNN belief model $N_p$, target network update frequency $Z$, batch size $D$, exploration parameter $\epsilon'$
**Result:** Robust Q network $Q_r$

1   Initialize replay buffer $\mathcal{B}$, robust Q network $Q_r = Q_v$, target Q network $Q' = Q_v$, observation history $S_{his}$, action history $A_{his}$;
2   **for** $t = 0,1,...,T$ **do**
3     Use PGD to find the best perturb state $\tilde{s}_t$ that minimizes $Q_r(s_t, \pi(\tilde{s}_t))$, where $\pi$ is derived from $Q_r$ by taking greedy action;
4     $M_t = M_t \cap B_\epsilon(\tilde{s}_t)$;
5     Choose an action based on belief $M_t$ and $Q_r$ using $\epsilon$-greedy:
      $a_t = \operatorname{argmax}_{a \in A} \min_{m \in M_t} Q_r(m,a)$ with probability $1 - \epsilon'$; otherwise $a_t$ is a random action;
6     Append $\tilde{s}_t$ and $a_t$ to $S_{his}$ and $A_{his}$ and use belief model $N_p(S_{his}, A_{his})$ to generate $M_{t+1}$;
7     Execute action $a_t$ in the environment and observe reward $R_t$ and next true state $s_{t+1}$;
8     **if** $s_{t+1}$ *is a terminal state* **then**
9       Reset $S_{his}$ and $A_{his}$
10     **end**
11     Store transition $\{s_t, a_t, R_t, s_{t+1}, M_t\}$ in $\mathcal{B}$;
12     Sample a random minibatch of size $D$ of transitions $\{s_i, a_i, R_i, s_{i+1}, M_i\}$ from $\mathcal{B}$;
13     Set $y_i = \begin{cases} R_i & \text{for terminal } s_{i+1} \\ R_i + \gamma \max_{a' \in A} \min_{m \in M_i} Q'(m, a') & \text{for non-terminal } s_{i+1} \end{cases}$
14     Perform a gradient descent step to minimize $Huber(\Sigma_i y_i - Q_r(s_i, a_i))$;
15     Update target network every $Z$ steps;
16   **end**

---

---

**Algorithm 5:** Belief-Enriched Pessimistic DQN (BP-DQN) Testing

**Data:** Trained robust Q network $Q_r$, PFRNN belief model $N_p$

1   Initialize observation history $S_{his}$ and action history $A_{his}$;

2   **for** *t = 0,1,...,T* **do**

3      Observe the perturbed state $\tilde{s}_t$;

4      **if** *t = 0* **then**

5         $M_0 = B_\epsilon(\tilde{s}_t)$;

6      **end**

7      Select an action based on belief $M_t$ and $Q_r$: $a_t = \text{argmax}_{a \in A} \min_{m \in M_t} Q_r(m, a)$;

8      Append $\tilde{s}_t$ and $a_t$ to $S_{his}$ and $A_{his}$ and use belief model $N_p(S_{his}, A_{his})$ to generate $M_{t+1}$;

9      Execute action $a_t$ in the environment;

10   **end**

---

**Algorithm 6:** Diffusion-Assisted Pessimistic DQN (DP-DQN) Training. We highlight the difference between our algorithm and the vanilla DQN algorithm in brown.

**Data:** Number of iterations $T$, trained vanilla Q network $Q_v$, diffusion belief model $N_d$, target network update frequency $Z$, batch size $D$, belief size $\kappa_d$, exploration parameter $\epsilon'$, noise level $\epsilon_\phi$

**Result:** Robust Q network $Q_r$

1   Initialize replay buffer $\mathcal{B}$, robust Q network $Q_r = Q_v$, target Q network $Q' = Q_v$;

2   **for** *t = 0,1,...,T* **do**

3      Use PGD to find the best perturb state $\tilde{s}_t$ that minimizes $Q_r(s_t, \pi(\tilde{s}_t))$, where $\pi$ is derived from $Q_r$ by taking greedy action;

4      Sample noise $\phi$ from Gaussian distribution $\mathcal{N}(0, \epsilon_\phi^2)$ pixel-wise with same dimension as $s_t$;

5      Use the diffusion belief model to generate belief $M_t = N_d(\tilde{s}_t + \phi)$ of size $\kappa_d$;

6      Select an actions based on belief $M_t$ and $Q_r$ using $\epsilon$-greedy:
     $a_t = \text{argmax}_{a \in A} \min_{m \in M_t} Q_r(m, a)$ with probability $1 - \epsilon'$; otherwise $a_t$ is a random action;

7      Execute action $a_t$ in environment and observe reward $R_t$ and next true state $s_{t+1}$;

8      Apply the reverse diffusion process to $s_t$ and $s_{t+1}$: $\hat{s}_t = N_d(s_t)$, $\hat{s}_{t+1} = N_d(s_{t+1})$;

9      Store transition $\{\hat{s}_t, a_t, R_t, \hat{s}_{t+1}, M_t\}$ in $\mathcal{B}$;

10     Sample a random minibatch of size $D$ of transitions $\{\hat{s}_i, a_i, R_i, \hat{s}_{i+1}, M_i\}$ from $\mathcal{B}$;

11     Set $y_i = \begin{cases} R_i & \text{for terminal } \hat{s}_{i+1} \\ R_i + \gamma \max_{a' \in A} \min_{m \in M_i} Q'(m, a') & \text{for non-terminal } \hat{s}_{i+1} \end{cases}$

12     Perform a gradient descent step to minimize $Huber(\Sigma_i y_i - Q_r(\hat{s}_i, a_i))$;

13     Update target network every $Z$ steps;

14   **end**

---

**Algorithm 7:** Diffusion-Assisted Pessimistic DQN (DP-DQN) Testing

**Data:** Trained robust Q network $Q_r$, diffusion belief model $N_d$, noise level $\epsilon_\phi$

1   **for** *t = 0,1,...,T* **do**

2      Observe the perturbed state $\tilde{s}_t$;

3      Sample noise $\phi$ from Gaussian distribution $\mathcal{N}(0, \epsilon_\phi^2)$ pixel-wise with same dimension as $s_t$;

4      Generate belief using the diffusion belief model $M_t = N_d(\tilde{s}_t + \phi)$;

5      Choose an action based on belief $M_t$ and $Q_r$: $a_t = \text{argmax}_{a \in A} \min_{m \in M_t} Q_r(m, a)$;

6      Execute action $a_t$ in the environment;

7   **end**

---

## F   EXPERIMENT DETAILS AND ADDITIONAL RESULTS

### F.1   EXPERIMENT SETUP JUSTIFICATION

Although both the BP-DQN and DP-DQN algorithms follow our idea of pessimistic Q-learning, the former is more appropriate for games with a discrete or a continuous but low-dimensional state space

such as the continuous Gridworld environment, while the latter is more appropriate for games with raw pixel input such as Atari Games.

In the Gridworld environment, state perturbations can manipulate the semantics of states by changing the coordinates of the agent. In this case, historical information can be utilized to generate beliefs about true states. Following this idea, BP-DQN uses the particle filter recurrent neural network (PF-RNN) method to predict true states. In principle, we can also use BP-DQN on the Atari environments to predict true states through historical data. In practice, however, it is computationally challenging to do so due to the high-dimensional state space ($84 \times 84$) of the Atari environments. Developing more efficient belief update techniques for large environments remains an active research direction.

On the other hand, state perturbations are injected pixel-wise in state-of-the-art attacks in Atari games. Consequently, they can barely change the semantics of true states Atari environments. In this case, historical information becomes less useful, and the diffusion model can effectively "purify" the perturbed states to recover the true states from high-dimensional image data. Although it is theoretically possible to use DP-DQN on the Gridworld environment, we conjecture that it is less effective than BP-DQN since it does not utilize historical data, which is crucial to recover true states when perturbations can change the semantic meaning of states as in the case of continuous Gridworld. In particular, we observe that the distributions of perturbed states and true states are very similar in this environment, making it difficult to learn a diffusion model that can map the perturbed states back to true states.

It is an interesting direction to develop strong perturbation attacks that can manipulate the semantics of true states for games with raw pixel input. As a countermeasure, we can potentially integrate diffusion-based state purification and belief-based history modeling to craft a stronger defense.

### F.2 EXPERIMENT SETUP

**Environments.** The continuous state Gridworld is modified from the grid maze environment in (Ma et al., 2020). We create a $10 \times 10$ map with walls inside. There are also gold and a bomb in the environment where the agent aims to find the gold and avoid the bomb. The state space is a tuple of two real numbers in $[0, 10] \times [0, 10]$ representing the coordinate of the agent. The initial state of the agent is randomized. The agent can move in 8 directions, which are up, up left, left, down left, down, down right, right, and up right. By taking an action, the agent moves a distance of 0.5 units in the direction they choose. For example, if the agent is currently positioned at $(x, y)$ and chooses to move upwards, the next state will be $(x, y + 0.5)$. If the agent chooses to move diagonally to the upper right, the next state will be $(x + 0.5/\sqrt{2}, y + 0.5/\sqrt{2})$. If the agent would collide with a wall by taking an action, it remains stationary at its current location during that step. The agent loses 1 point for each time step before the game ends and gains a reward of 200 points for reaching the gold and $-50$ points for reaching the bomb. The game terminates once the agent reaches the gold or bomb or spends 100 steps in the game. For Atari games, we choose Pong and Freeway provided by the OpenAI Gym (Brockman et al., 2016).

**Baselines.** We choose vanilla DQN (Mnih et al., 2015), SA-DQN (Zhang et al., 2020a) and WocaR-DQN (Liang et al., 2022) as defense baselines. We consider three commonly used attacks to evaluate the robustness of these algorithms: (1) PGD attack Zhang et al. (2020a), which aims to find a perturbed state $\tilde{s}$ that minimizes $Q(s, \pi(\tilde{s}))$ and we set PGD steps $\eta = 10$ for both training and testing usage; (2) MinBest attack (Huang et al., 2017), which aims to find a perturbed state $\tilde{s}$ that minimizes the probability of choosing the best action under $s$, with the probabilities of actions represented by a softmax of Q-values; and (3) PA-AD (Sun et al., 2021), which utilizes RL to find a (nearly) optimal attack policy. For each attack, we choose $\epsilon \in \{0.1, 0.5\}$ for the Gridworld environment and $\epsilon \in \{1/255, 3/255, 15/255\}$ for the Atari games. Natural rewards (without attacks) are reported using policies trained under $\epsilon = 0.1$ for continuous state Gridworld and $\epsilon = 1/255$ for Atari games.

**Training and Testing Details.** We use the same network structure as vanilla DQN (Mnih et al., 2015), which is also used in SA-DQN (Zhang et al., 2020a) and WocaR-DQN (Liang et al., 2022). We set all parameters as default in their papers when training both SA-DQN and WocaR-DQN. For training our pessimistic DQN algorithm with PF-RNN-based belief (called BP-DQN, see Algorithm 4 in Appendix E), we set $\kappa_p = |M_t| = 30$, i.e., the PF-RNN model will generate 30 belief states in each time step. For training our pessimistic DQN algorithm with diffusion (called DP-DQN, see

| Env | Parameter | PGD | | | MinBest | | | PA-AD | | |
|---|---|---|---|---|---|---|---|---|---|---|
| | | $\epsilon = 1/255$ | $\epsilon = 3/255$ | $\epsilon = 15/255$ | $\epsilon = 1/255$ | $\epsilon = 3/255$ | $\epsilon = 15/255$ | $\epsilon = 1/255$ | $\epsilon = 3/255$ | $\epsilon = 15/255$ |
| Pong | noise level $\epsilon_\phi$ | 3/255 | 5/255 | 10/255 | 3/255 | 1/255 | 1/255 | 2/255 | 3/255 | 10/255 |
| | reverse step $k$ | 1 | 1 | 4 | 1 | 1 | 1 | 2 | 2 | 3 |
| | sampler step | 128 | 64 | 128 | 128 | 128 | 32 | 64 | 64 | 32 |
| Freeway | noise level $\epsilon_\phi$ | 3/255 | 3/255 | 3/255 | 3/255 | 3/255 | 3/255 | 3/255 | 3/255 | 3/255 |
| | reverse step $k$ | 2 | 4 | 4 | 2 | 4 | 4 | 2 | 4 | 4 |
| | sampler step | 64 | 64 | 64 | 64 | 64 | 64 | 64 | 64 | 64 |

Table 4: Parameters Used to Test DP-DQN-F

Algorithm 6 in Appendix E), we set $\kappa_d = |M_t| = 4$, that is, the diffusion model generates 4 purified belief states from a perturbed state. We consider two variants of DP-DQN, namely, DP-DQN-O and DP-DQN-F, which utilize DDPM and Progressive Distillation as the diffusion model, respectively. For DP-DQN-O, we set the number of reverse steps to $k = 10$ for $\epsilon = 1/255$ or $3/255$ and $k = 30$ for $\epsilon = 15/255$, and do not add noise $\phi$ when training and testing DP-DQN-O. For DP-DQN-F, we set $k = 1$, sampler step to 64, and add random noise with $\epsilon_\phi = 5/255$ when training DP-DQN-F. We report the parameters used when testing DP-DQN-F in Table 4. We sample $C = C' = 30$ trajectories to train PF-RNN and diffusion models. All other parameters are set as default for training the PF-RNN and diffusion models. For all other baselines, we train 1 million frames for the continuous Gridworld environment and 6 million frames for the Atari games. For our methods, we take the pre-trained vanilla DQN model, and train our method for another 1 million frames. All training and testing are done on a machine equipped with an i9-12900KF CPU and a single RTX 3090 GPU. For each environment, all RL policies are tested in 10 randomized environments with means and variances reported.

## F.3 MORE EXPERIMENT RESULTS

### F.3.1 MORE BASELINE RESULTS

Tables 5a and 5b give the complete results for the continuous space Gridworld and the two Atari games, where we include another baseline called Radial-DQN (Oikarinen et al., 2021), which adds an adversarial loss term to the nominal loss of regular DRL in order to gain robustness. We find that Radial-DQN fails to learn a reasonable policy in continuous Gridworld as other regularization-based methods. However, Radial-DQN performs well under a small attack budget in Atari games but still fails to respond when the attack budget is high. Our Radial-DQN results for the Atari games were obtained using the pre-trained models (Oikarinen et al., 2021), which might explain why the results are better than those reported in (Liang et al., 2022).

| Environment | Model | Natural Reward | PGD | | MinBest | | PA-AD | |
|---|---|---|---|---|---|---|---|---|
| | | | $\epsilon = 0.1$ | $\epsilon = 0.5$ | $\epsilon = 0.1$ | $\epsilon = 0.5$ | $\epsilon = 0.1$ | $\epsilon = 0.5$ |
| GridWorld Continous | DQN | $156.5 \pm 90.2$ | $128 \pm 118$ | $-53 \pm 86$ | $98.2 \pm 137$ | $98.2 \pm 137$ | $-10.7 \pm 136$ | $-35.9 \pm 118$ |
| | SA-DQN | $20.8 \pm 140$ | $46 \pm 142$ | $-100 \pm 0$ | $-5.8 \pm 131$ | $-100 \pm 0$ | $-97.5 \pm 13.6$ | $-67.8 \pm 78.3$ |
| | WocaR-DQN | $-63 \pm 88$ | $-100 \pm 0$ | $-63.2 \pm 88$ | $-100 \pm 0$ | $-63.2 \pm 88$ | $-100 \pm 0$ | $-63.2 \pm 88$ |
| | Radial-DQN | $-100 \pm 0$ | $-96.1 \pm 12.3$ | $-96.1 \pm 12.3$ | $-100 \pm 0$ | $-100 \pm 0$ | $-100 \pm 0$ | $-100 \pm 0$ |
| | BP-DQN (Ours) | $163 \pm 26$ | $165 \pm 29$ | $176 \pm 16$ | $147 \pm 88$ | $114 \pm 114$ | $171.9 \pm 17$ | $177.2 \pm 10.6$ |

(a) Continuous Gridworld Results

| Env | Model | Natural Reward | PGD | | | MinBest | | | PA-AD | | |
|---|---|---|---|---|---|---|---|---|---|---|---|
| | | | $\epsilon = 1/255$ | $\epsilon = 3/255$ | $\epsilon = 15/255$ | $\epsilon = 1/255$ | $\epsilon = 3/255$ | $\epsilon = 15/255$ | $\epsilon = 1/255$ | $\epsilon = 3/255$ | $\epsilon = 15/255$ |
| Pong | DQN | $21 \pm 0$ | $-21 \pm 0$ | $-21 \pm 0$ | $-21 \pm 0$ | $-21 \pm 0$ | $-21 \pm 0$ | $-21 \pm 0$ | $-18.2 \pm 2.3$ | $-19 \pm 2.2$ | $-21 \pm 0$ |
| | SA-DQN | $21 \pm 0$ | $21 \pm 0$ | $21 \pm 0$ | $-20.8 \pm 0.4$ | $21 \pm 0$ | $21 \pm 0$ | $-21 \pm 0$ | $21 \pm 0$ | $18.7 \pm 2.6$ | $-20 \pm 0$ |
| | WocaR-DQN | $21 \pm 0$ | $21 \pm 0$ | $21 \pm 0$ | $-21 \pm 0$ | $21 \pm 0$ | $21 \pm 0$ | $-21 \pm 0$ | $21 \pm 0$ | $19.7 \pm 2.4$ | $-21 \pm 0$ |
| | Radial-DQN | $21 \pm 0$ | $21 \pm 0$ | $21 \pm 0$ | $-21 \pm 0$ | $21 \pm 0$ | $21 \pm 0$ | $-21 \pm 0$ | $21 \pm 0$ | $21 \pm 0$ | $-19 \pm 0$ |
| | DP-DQN-O (Ours) | $19.9 \pm 0.3$ | $19.9 \pm 0.3$ | $19.8 \pm 0.4$ | $19.7 \pm 0.5$ | $19.9 \pm 0.3$ | $19.9 \pm 0.3$ | $19.3 \pm 0.8$ | $19.9 \pm 0.3$ | $19.9 \pm 0.3$ | $19.3 \pm 0.8$ |
| | DP-DQN-F (Ours) | $20.8 \pm 0.4$ | $20.4 \pm 0.9$ | $20.4 \pm 0.9$ | $18.3 \pm 1.9$ | $20.6 \pm 0.9$ | $20.4 \pm 0.8$ | $21.0 \pm 0.0$ | $18.6 \pm 2.5$ | $20.0 \pm 1$ | $18.2 \pm 1.8$ |
| Freeway | DQN | $34 \pm 0.1$ | $0 \pm 0$ | $0 \pm 0$ | $0 \pm 0$ | $0 \pm 0$ | $0 \pm 0$ | $0 \pm 0$ | $0 \pm 0$ | $0 \pm 0$ | $0 \pm 0$ |
| | SA-DQN | $30 \pm 0$ | $30 \pm 0$ | $30 \pm 0$ | $0 \pm 0$ | $27.2 \pm 3.4$ | $18.3 \pm 3.0$ | $0 \pm 0$ | $20.1 \pm 4.0$ | $9.5 \pm 3.8$ | $0 \pm 0$ |
| | WocaR-DQN | $31.2 \pm 0.4$ | $31.2 \pm 0.5$ | $31.4 \pm 0.3$ | $21.6 \pm 1$ | $29.6 \pm 2.5$ | $19.8 \pm 3.8$ | $21.6 \pm 1$ | $24.9 \pm 3.7$ | $12.3 \pm 3.2$ | $21.6 \pm 1$ |
| | Radial-DQN | $33.4 \pm 0.5$ | $33.4 \pm 0.5$ | $33.4 \pm 0.5$ | $21.6 \pm 1$ | $33.4 \pm 0.5$ | $32.8 \pm 0.8$ | $21.6 \pm 1$ | $33.4 \pm 0.5$ | $33.4 \pm 0.5$ | $21.6 \pm 1$ |
| | DP-DQN-O (Ours) | $28.8 \pm 1.1$ | $29.1 \pm 1.1$ | $29 \pm 0.9$ | $28.9 \pm 0.7$ | $29.2 \pm 1.0$ | $28.5 \pm 1.2$ | $28.6 \pm 1.3$ | $28.6 \pm 1.2$ | $28.3 \pm 1$ | $28.8 \pm 1.3$ |
| | DP-DQN-F (Ours) | $31.2 \pm 1$ | $30.0 \pm 0.9$ | $30.1 \pm 1$ | $30.7 \pm 1.2$ | $30.2 \pm 1.3$ | $30.6 \pm 1.4$ | $29.4 \pm 1.2$ | $30.8 \pm 1$ | $31.4 \pm 0.8$ | $28.9 \pm 1.1$ |

(b) Atari Games Results

Table 5: Experiment Results. We show the average episode rewards ± standard deviation over 10 episodes for our methods and three baselines. The results for our methods are highlighted in gray.

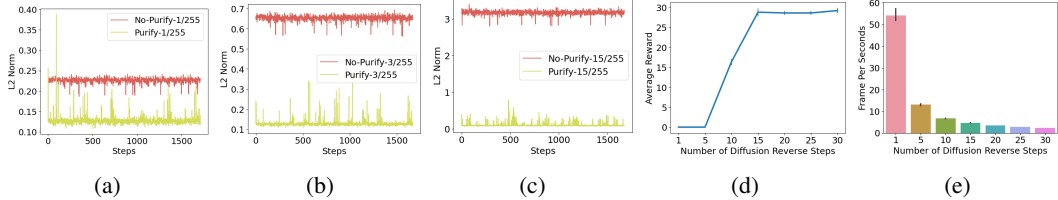

Figure 5: a), b) and c) show the $l_2$ distance between perturbed states and original states before and after purification under different attack budgets in the Pong environment using DDPM. d) shows the performance of DP-DQN-O under different diffusion steps in the Freeway environment under PGD attack with $\epsilon = 15/255$. e) shows the testing stage speed of DP-DQN-O (measured by the number of frames processed per second) under different diffusion steps in the Freeway environment.

| Environment | Model | Training (hours) | Testing (FPS) | | Environment | Model | Training (hours) | Testing (FPS) |
|---|---|---|---|---|---|---|---|---|
| **GridWorld Continous** | SA-DQN | 3 | 607 | | **Pong** | SA-DQN | 38 | 502 |
| | WocaR-DQN | 3.5 | 721 | | | WocaR-DQN | 50 | 635 |
| | BP-DQN (Ours) | 0.6+1.5+7 | 192 | | | DP-DQN-O (Ours) | 1.5+18+30 | 6.6 |
| | | | | | | DP-DQN-F (Ours) | 1+18+24 | 93 |

Table 6: Training and Testing Time Comparison. The training of our methods contains three parts: a) training the PF-RNN or diffusion model, b) training a vanilla DQN policy without attacks, and c) training a robust policy using BP-DQN, DP-DQN-O, or DP-DQN-F.

### F.3.2 MORE ABLATION STUDY RESULTS

**Diffusion Effects.** In Figures 5a-5c, we visualize the effect of DDPM-based diffusion by recording the $l_2$ distance between a true state and the perturbed state and that between a purified true state and the purified perturbed state. For all three levels of attack budgets, our diffusion model successfully shrinks the gap between true states and perturbed states.

**Performance vs. Running Time in DP-DQN.** We study the impact of different diffusion steps $k$ on average return of DP-DQN-O in Figure 5d and their testing stage running time in Figure 5e. Figure 5d shows the performance under different diffusion steps of our method in the Freeway environment under PGD attack with budget $\epsilon = 15/255$. It shows that we need enough diffusion steps to gain good robustness, and more diffusion steps do not harm the return but do incur extra overhead, as shown in Figure 5e, where we plot the testing stage running time in Frame Per Second (FPS). As the number of diffusion steps increases, the running time of our method also increases, as expected.

On the other hand, as DP-DQN-F uses a distilled sampler, it can decrease the reverse sample step $k$ to as small as 1, which greatly reduces the testing time as reported in Table 6. We report the performance results of DP-DQN-F in Table 1b and we find that DP-DQN-F improves DP-DQN-O under small perturbations, but suffers performance loss in Pong under PA-AD attack and large perturbations. Further, DP-DQN-F has a larger standard deviation than DP-DQN in the Pong environment, indicating that DP-DQN-F is less stable than DP-DQN-O in Pong. We conjecture that the lower sample quality introduced by Progressive Distillation causes less stable performance and performance loss under PA-AD attack compared to DP-DQN-O that utilizes DDPM.

**Training and Testing Overhead.** Table 6 compares the training and test-stage overhead of SA-DQN, WocaR-DQN, and our methods. Notice that the training of our methods consists of three parts: a) training the PF-RNN belief model or the diffusion model, b) training a vanilla DQN policy without attacks, and c) training a robust policy using BP-DQN, DP-DQN-O or DP-DQN-F. In the continuous state Gridworld environment, our method takes around 9 hours to finish training, which is higher than SA-DQN and WocaR-DQN. But our method significantly outperforms these two baselines as shown in Table 1a. In the Atari Pong game, our method takes about 50 hours to train, which is comparable to WocaR-DQN but slower than SA-DQN. In terms of running time at the test stage, we calculate the FPS of each method and report the average FPS over 5 testing episodes. In the continuous Gridworld environment, our method is slower but comparable to SA-DQN and WocaR-DQN due to the belief update and $\mathrm{maximin}$ search. However, in the Atari Pong game, our DP-DQN-O method is much slower than both SA-DQN and WocaR-DQN. This is mainly due to the use of a large diffusion model

| Environment | Model | Natural Reward | PGD | | MinBest | | PA-AD | |
|---|---|---|---|---|---|---|---|---|
| | | | $\epsilon = 0.1$ | $\epsilon = 0.5$ | $\epsilon = 0.1$ | $\epsilon = 0.5$ | $\epsilon = 0.1$ | $\epsilon = 0.5$ |
| **GridWorld Continous (Clean)** | BP-DQN | $163 \pm 26$ | $165 \pm 29$ | $176 \pm 16$ | $147 \pm 88$ | $114 \pm 114$ | $171.9 \pm 17$ | $177.2 \pm 10.6$ |
| **GridWorld Continous** | BP-DQN | $102 \pm 110$ | $101.5 \pm 109$ | $56 \pm 135$ | $79.3 \pm 125$ | $30.7 \pm 138$ | $78.6 \pm 125$ | $13.1 \pm 140$ |

(a) Continuous Gridworld Results

| Env | Model | Natural Reward | PGD | | | MinBest | | | PA-AD | | |
|---|---|---|---|---|---|---|---|---|---|---|---|
| | | | $\epsilon = 1/255$ | $\epsilon = 3/255$ | $\epsilon = 15/255$ | $\epsilon = 1/255$ | $\epsilon = 3/255$ | $\epsilon = 15/255$ | $\epsilon = 1/255$ | $\epsilon = 3/255$ | $\epsilon = 15/255$ |
| **Freeway (Clean)** | DP-DQN-F | $31.2 \pm 1$ | $30.0 \pm 0.9$ | $30.1 \pm 1$ | $30.7 \pm 1.2$ | $30.2 \pm 1.3$ | $30.6 \pm 1.4$ | $29.4 \pm 1.2$ | $30.8 \pm 1$ | $31.4 \pm 0.8$ | $28.9 \pm 1.1$ |
| **Freeway** | DP-DQN-F | $29.4 \pm 0.9$ | $29.0 \pm 1.2$ | $28.8 \pm 1.5$ | $29.0 \pm 1.4$ | $28.8 \pm 0.4$ | $29.6 \pm 1.1$ | $29 \pm 1.6$ | $29.2 \pm 0.4$ | $28.2 \pm 1.8$ | $23.6 \pm 1.1$ |

(b) Atari Games Results

Table 7: Noisy Environment Results. We show the average episode rewards $\pm$ standard deviation over 10 episodes for our methods trained in a noisy environment.

in our method. However, our DP-DQN-F method is around 13 times faster than DP-DQN-O and is comparable to SA-DQN and WocaR-DQN.

**Noisy Environment Results.** Existing studies (including all the baselines we used) on robust RL against adversarial state perturbations commonly assume that the agent has access to a clean environment at the training stage. This is mainly because an RL agent may suffer from a significant amount of loss if it has to explore a poisoned environment on the fly, which is infeasible for security-sensitive applications. However, it is an important direction to study the more challenging setting where the agent has access to a noisy environment only, and we have conducted a preliminary investigation. We assumed that the agent could access a small amount of clean data (30 episodes of clean trajectories were used in the paper) to train a belief model or a diffusion model. Note that to learn the diffusion model, we only need a set of clean states but not necessarily complete trajectories. We then trained an RL policy in a poisoned environment by modifying the BP-DQN and DP-DQN algorithms as follows. For BP-DQN, we changed line 11 of Algorithm 4 (in Appendix E) to store the average of belief states (obtained from the perturbed states) instead of the true states in the replay buffer. For DP-DQN, we changed lines 8 and 9 of Algorithm 6 (in Appendix E) to apply the diffusion-based purification on perturbed states instead of true states. We trained these policies against PGD-poisoned continuous Gridworld and Freeway environments, with an attack budget of 0.1 and 1/255, respectively, and then tested them in the same environment. The results are as shown in Table 7a and 7b. Although our algorithms suffer from a small performance loss when the training environment is noisy, they can still achieve a high level of robustness under all test cases.

