# OpenReview forum: "Belief-Enriched Pessimistic Q-Learning against Adversarial State Perturbations"
_ICLR.cc/2024/Conference — ICLR 2024 poster_

### Official Review · Reviewer_cDHy · 2023-10-30

**Soundness:** 3 good
**Presentation:** 3 good
**Contribution:** 3 good
**Rating:** 6
**Confidence:** 2

**Summary:**

This paper addresses the vulnerability of reinforcement learning agents to adversarial attacks by manipulating state observations. The authors propose a algorithm focusing on developing a pessimistic policy that accounts for uncertainties in state information. This is supplemented by belief state inference and diffusion-based state purification techniques.

**Strengths:**

Addressed an important problem in Reinforcement Learning, providing fresh perspective and insights.
The experimental design and methodology are well-constructed.

**Weaknesses:**

The innovative aspects presented over WocaR-DQN appear to be incremental in nature.

**Questions:**

How extensive or generalizable are the empirical results presented? Given that the study focuses on the Continuous Gridworld, which is relatively simple, and only includes two Atari games.

Considering that Atari game screens are depicted using 8-bit (256 levels) RGB, yet typically select colors from a narrower palette, how effective are the different attack budgets (15/255, 3/255, 1/255) tested in this study? Specifically, can these perturbations alter the colors enough to cause confusion with other colors in the game's limited palette? If not, could the observed robustness of the learning method simply be a consequence of learning to filter out specific colors?

Is the proposed method also applicable to environments like Mujoco or Mountain Car, where the input variables are more continuous and less discrete than those in Atari games?

---

> ### Author Response · Authors · 2023-11-16
>
> We appreciate your constructive comments and feedback. Now we will explain your concerns and questions point by point in the following.
>
> Q1: The innovative aspects presented over WocaR-DQN appear to be incremental in nature.
>
> A1: Our method differs from WocaR-DQN in three important aspects. First, we adopt the maximin framework as a principled approach for achieving robustness, which is applied at both training and test stages using the perturbed states as input. In contrast, WocaR-DQN uses a regularization term and an estimation of the worst-case loss in the loss function to gain robustness. While the former is insufficient in the face of strong attacks, the latter is estimated using a separate neural network, suffering from the gap between training and testing. Second, we incorporate belief modeling into robust decision making, which is crucial for combating strong attacks, but has not been considered in WocaR-DQN. Third, we introduce diffusion-based state purification for environments with raw-pixel input, which allows us to train a single policy against different attack budgets. In contrast, a separate policy is required in WocaR-DQN.
>
> Q2: We only test on two atari games.
>
> A2: We have conducted a new experiment and tested our methods on the Bankheist environment that is widely used in other baselines. However, due to the recent upgrade of the Gym package, all pre-trained models provided by other baselines failed to work anymore. Thus, we retrained all other baselines ourselves, which resulted in performances different from those reported in the original papers. The result of Bankheist is shown below.
> |     Env    |      Model     |  Natural  Reward |         PGD        |                    |                     |       MinBest      |                    |                     |        PA-AD       |                    |                     |
> |:----------:|:--------------:|:----------------:|:------------------:|:------------------:|:-------------------:|:------------------:|:------------------:|:-------------------:|:------------------:|:------------------:|:-------------------:|
> |            |                |                  | $\epsilon = 1/255$ | $\epsilon = 3/255$ | $\epsilon = 15/255$ | $\epsilon = 1/255$ | $\epsilon = 3/255$ | $\epsilon = 15/255$ | $\epsilon = 1/255$ | $\epsilon = 3/255$ | $\epsilon = 15/255$ |
> | Bank Heist |       DQN      |    $680 \pm 0$   |      $0 \pm 0$     |      $0 \pm 0$     |      $0 \pm 0$      |      $0 \pm 0$     |      $0 \pm 0$     |      $0 \pm 0$      |     $10 \pm 0$     |     $10 \pm 0$     |      $0 \pm 0$      |
> | Bank Heist |     SA_DQN     |    $690 \pm 0$   |     $690 \pm 0$    |      $0 \pm 0$     |      $10 \pm 0$     |     $690 \pm 0$    |      $0 \pm 0$     |      $0 \pm 0$      |     $670 \pm 0$    |     $630 \pm 0$    |      $60 \pm 0$     |
> | Bank Heist |    WocaR_DQN   |    $680 \pm 0$   |    $354 \pm 309$   |      $0 \pm 0$     |      $0 \pm 0$      |     $430 \pm 0$    |      $0 \pm 0$     |      $0 \pm 0$      |     $580 \pm 0$    |     $120 \pm 0$    |      $0 \pm 0$      |
> | Bank Heist | DP_DQN_O(Ours) |   $680 \pm 10$   |    $684 \pm 13$    |    $638 \pm 50$    |     $108 \pm 28$    |    $700 \pm 10$    |    $678 \pm 15$    |     $85 \pm 5.4$    |    $708 \pm 23$    |    $686 \pm 18$    |    $192 \pm 107$    |
> | Bank Heist | DP_DQN_F(Ours) |  $694 \pm 11.4$  |   $688 \pm 13.0$   |   $682 \pm 13.0$   |    $472 \pm 132$    |    $686 \pm 5.4$   |     $700 \pm 7$    |    $244 \pm 106$    |    $700 \pm 12$    |    $702 \pm 4.4$   |    $490 \pm 50.5$   |
>
> Our DP-DQN-O and DP-DQN-F methods outperform other baselines in the BankHeist environment, although they suffer from a notable performance loss when the attack budget is 15/255 due to the complexity of the game. Also, DP-DQN-F shows better robustness than DP-DQN-O when $\epsilon=15/255$.
>
> Answers for Q3 and Q4 are in the next comment due to the characters limitation.

---

> > ### Author Response · Authors · 2023-11-16
> >
> > Q3: How effective are the different attack budgets (15/255, 3/255, 1/255) when the game screens are depicted using 8-bit (256 levels) RGB. Could the observed robustness of the learning method simply be a consequence of learning to filter out specific colors?
> >
> > A3: We would like to thank the reviewer for sharing this interesting idea with us. The color filtering approach is in spirit similar to our diffusion-based purification, as both approaches aim to convert the invalid perturbed states back to the closest valid states. We remark that all the existing attacks and defenses including ours use normalized grayscale images as input, where pixel values are in the range of [0,1] and the attack budget (1/255, 3/255, 15/255) is defined with respect to this range. This is also a common technique for training RL algorithms in Atari environments. Although it is certainly possible to launch attacks against the RGB images, building a perfect color filter can be hard as it requires domain knowledge to identify all the valid colors. Further, there are Atari environments where the game screens may involve very similar colors, especially during the “flickering” phases of the games. Examples include Bankheist, Assault, Battle Zone, Demon Attack, and Haunted House. For environments like these, it is not easy to filter out specific colors to remove the impact of perturbations.
> >
> > Q4: Extension to policy-based RL to apply on Mountain Car and Mojuco environments.
> >
> > A4: We have focused on value-based methods in this work as a starting point to explore the robustness given by our pessimistic learning and belief approximation methods and bound the performance loss on a Q-learning algorithm. Our BP-DQN and DP-DQN algorithms cannot be directly applied to Mountain Car or Mojuco environments with continuous actions. However, we envision that some of the key ideas of the paper, including belief modeling and state purification, can be easily adapted to policy-based methods such as PPO and A2C, while more thoughts are needed for incorporating pessimistic learning into these algorithms.
> >
> > We hope that our explanations address your major concerns. If there are still concerns or questions, we would be happy to hear and discuss them.

---

> ### Author Response · Authors · 2023-11-22
> **Does our response address your concerns?**
>
> Dear reviewer cDHy,
>
> As the stage of the review discussion is ending soon, we would like to kindly ask you to review our response and consider making adjustments to the scores.
>
> Sincerely,
>
> Paper3965 Authors

---

### Official Review · Reviewer_GsDy · 2023-10-31

**Soundness:** 3 good
**Presentation:** 3 good
**Contribution:** 2 fair
**Rating:** 6
**Confidence:** 2

**Summary:**

The paper introduces a novel RL algorithm that aims to protect RL agents from adversarial state perturbations. The authors propose a pessimistic DQN algorithm that takes into consideration both the worst-case scenarios and belief about the true states. The algorithm also features a diffusion-based state purification method for applications like Atari games. The paper shows empirical results demonstrating that their approach significantly outperforms existing solutions in robustness against strong adversarial attacks, while maintaining comparable computational complexity.

**Strengths:**

1. The empirical results show performance improvement under strong attacks compared to baseline methods. The algorithm works well for both simplistic environments and more complex scenarios like Atari games with raw pixel input.
2. The algorithm's computational overhead is comparable to existing methods that use regularization terms.

**Weaknesses:**

1. The algorithm assumes access to a clean environment during training, which may not always be the case in real-world applications.
2. While the diffusion model adds robustness, it also adds computational overhead, potentially making it slower at test time.

**Questions:**

1. How sensitive is your algorithm to the choice of hyperparameters?
2. Given the requirement for a clean training environment, how would your method perform in a scenario where such an environment is not readily available?
3. The diffusion model increases computational complexity during the test stage. Are there ways to optimize this without compromising the robustness?
4. Why PA-AD is not evaluated on continuous gridworld?

---

> ### Author Response · Authors · 2023-11-16
>
> We appreciate your constructive comments and feedback. Now we will explain your concerns and questions point by point in the following.
>
> Q1: How sensitive our algorithm to the choice of hyperparameters?
>
> A1: We did not fine-tune most of the hyperparameters. For training the RL policies, we used the same default hyperparameters (e.g. learning rate, batch size, buffer size) as SA-MDP. For training the belief model, we took the default hyperparameters from the PF-RNN paper. For training the diffusion models, we used the default parameters in the DDPM and Progressive Diffusion papers, except for the number of reverse steps, which was significantly reduced to adapt the models to adversarial perturbations. As discussed in the second question proposed by Reviewer QHWx and Appendix F.3.2, our DP-DPN-O method is less sensitive to the number of reverse steps during testing, while our DP-DQN-F method is more sensitive to it.
>
> Q2: How would our method perform in a scenario where such an environment is not readily available?
>
> A2: We have added a general response to this question.
>
> Q3: PA-AD is not evaluated on continuous gridworld.
>
> A3:  Following the reviewer’s suggestion, we have adapted PA-AD to our continuous Gridworld environment and tested our methods and other baselines. The results are as follows:
> |  Env |  Model | PA-AD | PA-AD |
> |:---:|:---:|:---:|:---:|
> |  |  | $\epsilon = 0.1$ | $\epsilon = 0.5$ |
> | Continuous Gridworld | DQN | $-10.7 \pm 136$ | $-35.9 \pm 118 $ |
> | Continuous Gridworld | SA_DQN | $-97.5 \pm 13.6$ | $-67.8 \pm 78.3$ |
> | Continuous Gridworld | WocaR_DQN | $-100 \pm 0$ | $-63.2 \pm 88.6$ |
> | Continuous Gridworld | BP_DQN | $171.9 \pm 17$ | $177.2 \pm 10.6$ |
>
> Similar to other attack baselines reported in our paper, our BP-DQN method achieves the best robustness under the PA-AD attack.
>
> Q4: Computational overhead brought by diffusion model.
>
> A4:  We admit that using a diffusion model introduces additional overhead to our methods, especially for the DP-DQN-O method where the DDPM model is used. In this work, we considered a fast diffusion method called Progressive Diffusion in our DP-DQN-F method. As shown in Table 2(b), DP-DQN-F has significantly lower training and testing overhead than DP-DQN-O. To further speed up state purification, one possibility is to change the U-net structure used by the diffusion models to accelerate the reverse process. Another direction is to utilize other generative models such as GANs and autoencoders for state purification, which may not be as effective as diffusion-based methods but incur less computational overhead.
>
> We hope that our explanations address your major concerns. If there are still concerns or questions, we would be happy to hear and discuss them.

---

> ### Author Response · Authors · 2023-11-22
> **Does our response address your concerns?**
>
> Dear reviewer GsDy,
>
> As the stage of the review discussion is ending soon, we would like to kindly ask you to review our response and consider making adjustments to the scores.
>
> Sincerely,
>
> Paper3965 Authors

---

> > ### Comment · Reviewer_GsDy · 2023-11-22
> >
> > Thank you to the authors for the clarification and the additional experiments. My opinion remains in favor of accepting the paper.

---

### Official Review · Reviewer_achV · 2023-10-31

**Soundness:** 2 fair
**Presentation:** 3 good
**Contribution:** 3 good
**Rating:** 5
**Confidence:** 4

**Summary:**

The authors study the problem of defense in presence of perceived state attack in Reinforcement learning and propose a method to approximately solve the Stackelberg equilibrium between the agents and the adversary. Their method involves solving pessimistic Q-learning and estimating belief state of the agent and using them for state purification. They propose two algorithms called BP-DQN and DP-DQN to defend against adversarial attacks.

**Strengths:**

1. Proposes a novel method to defend against adversarial attacks in RL by combining pessimistic Q-learning with belief state estimation and state purification objective.

2. Authors provided theoretical results to compare the policy found by their algorithm to the optimal policy.

3. Presented interesting empirical experiments to demonstrate effectiveness of their algorithm on several examples.

**Weaknesses:**

1. It has been shown that Stackelberg Equilibrium as defined in Definition 2 need not always exist, refer to theorem 4.3 [https://arxiv.org/pdf/2212.02705.pdf](https://arxiv.org/pdf/2212.02705.pdf). So, finding an approximate solution for them is meaningless. However, non-existence of Stackelberg equilibrium is a worst case phenomenon. Authors should incorporate this in their paper.

2. It will be great if authors can include a short paragraph before section 3.2 discussing a big picture of their strategies before diving into each one of them. It would also help to include some mathematical details in each section.

3. Abstract wrongly mentions that past methods either regularize or retrain the policy. However, methods like Bharti et.al. just purify the states directly.

**Questions:**

1. It is well known that defense against perceived state attack requires solving a partially observable MDP which is a hard problem to solve in general. Could you clarify how your method is able to avoid these hardness issues?

---

> ### Author Response · Authors · 2023-11-16
>
> We appreciate your constructive comments and feedback. Now we will explain your concerns and questions point by point in the following.
>
> Q1: It has been shown that Stackelberg Equilibrium as defined in Definition 2 need not always exist, refer to theorem 4.3 https://arxiv.org/pdf/2212.02705.pdf. So, finding an approximate solution for them is meaningless. However, non-existence of Stackelberg equilibrium is a worst case phenomenon. Authors should incorporate this in their paper.
>
> A1: We are aware of the negative result the reviewer mentioned. Please see the discussion below Definition 2 in Section 3, where we have cited the SA-MAP paper where this result was originally proved (Theorem 4 in the SA-MDP paper). We are also aware of the paper the reviewer mentioned and have cited its journal version in the Related Work section (Appendix B.1). The negative result implies that a policy that obtains the best possible value (against the strongest attack) across all states does not exist in general, but it does not exclude the possibility of finding an approximate solution that is close to the optimal state-wise. Although we were not able to identify a policy that obtains an approximate Stackelberg Equilibrium in the standard sense, we showed that our pessimistic Q learning algorithm (Algorithm 1) obtains a value function that is close to the optimal Q function without attacks (Theorem 1) under common assumptions.
>
> Q2: It will be great if authors can include a short paragraph before section 3.2 discussing a big picture of their strategies before diving into each one of them.
>
> A2: We present our basic solution framework, pessimistic Q-learning, in Section 3.3. Our approach derives maximin actions from the Q-function using perturbed states as input to safeguard against the agent’s uncertainty about true states. Notably, this approach is applied at both training and test stages, thus bridging the gap between the two as in previous work. We then improve the algorithm by incorporating the agent’s belief about true states into action selection in Section 3.4. To obtain a practical solution in high-dimensional state spaces, we present our BP-DQN algorithm in this section, which incorporates the above ideas into the classic Deep Q-Network (DQN) algorithm to obtain pessimistic DQN with belief states approximated using the PF-RNN method. To further scale our method to environments with raw-pixel input, we present another algorithm, DP-DQN, in Section 3.5, where we incorporate a diffusion-based purification scheme into pessimistic DQN to purify the pixel-wise perturbations.
>
> Q3: Abstract wrongly mentions that past methods either regularize or retrain the policy. However, methods like Bharti et.al. just purify the states directly.
>
> A3: We would like to thank the reviewer for pointing out the work beyond regularization-based methods and alternating training methods. We are aware of the purification-based methods, including the work by Bharti et al., and have provided a detailed comparison between our method and Bharti et al. in Appendix B.3. In particular, our method has two important components not considered in Bharti et al., namely, the maximin formulation and belief update. There also exist studies that detect if a state is perturbed or not at the test stage, e.g., Xiong et al., which uses an autoencoder-based method to detect perturbations, as we mentioned in Section 3.3 and Appendix B.1. We did not mention them in the abstract because Bharti et al. focuses on the backdoor attack rather than test-stage state perturbation we considered, while Xiong et al. focuses on test stage detection given a pre-trained policy.
>
> Q4: How do our methods avoid the hardness in POMDP?
>
> A4:  We first note that while training a robust agent against adversarial state perturbations is closely related to POMDPs, they are also fundamentally different. In POMDPs, the agent’s observation is derived from a predetermined observation function as part of the environment. In contrast, the agent’s observation is determined by the attacker’s policy in adversarial state perturbations, which might be non-stationary or even adaptive, making the problem even harder.
>
> In this work, we have exploited two key observations to bypass this obstacle. First, the attacker has a bounded budget to avoid detection, where the perturbed state must land in the \epsilon ball centered at the true state so that the agent could utilize historical information to generate a relatively accurate belief about true states, which might not be the case in general POMDPs. Further, for environments with raw-pixel input, existing attacks generate perturbations in the form of pixel-wise noise, which could be purified with the help of a diffusion model to generate a belief about the true state. These ideas together with our pessimistic training framework ensure high robustness against strong state perturbation attacks.
>
> We hope that our explanations address your major concerns.

---

> ### Author Response · Authors · 2023-11-22
> **Does our response address your concerns?**
>
> Dear reviewer achV,
>
> As the stage of the review discussion is ending soon, we would like to kindly ask you to review our response and consider making adjustments to the scores.
>
> Sincerely,
>
> Paper3965 Authors

---

> ### Comment · Reviewer_achV · 2023-11-23
>
> Dear authors,
> Thank you for your response! I am not very satisfied with the response to Q1(it is still not clear to me how can your algorithm provide an approximate solution to the problem which admits no solution? The notion of "approximate solution" being used does not seem to be a standard one) and also Q4(In my opinion just by constraining the attacker in $\epsilon$-ball radius would not permit the defender to recover back true states even if uses the historical information. If it can indeed do so, is there some threshold on $\epsilon$ above which defense is not possible and does your theory say something about it - because for large epsilon one can always change any state to any other state?). Hence, I would maintain my score for now.

---

> ### Author Response · Authors · 2023-11-23
> **Further explainations for Q1 and Q4**
>
> Thank you for carefully reading our response and we are glad to further clarify your concerns.
>
> For Q1, an accurate definition of approximate Stackelberg Equilibrium is as follows. First, as shown in Theorem 4.11 of https://arxiv.org/pdf/2212.02705.pdf, when the initial state distribution $Pr(s_0)$ is known, there is an agent policy $\pi^*$ that maximizes the worst-case **expected** state value against the optimal state perturbation attack, that is
>
> $\forall \pi$, $\Sigma_{s_0\in S} Pr(s_0) V_{\pi^*\circ\omega_{\pi^*}}(s_0) \ge \Sigma_{s_0\in S} Pr(s_0) V_{\pi\circ\omega_{\pi}}(s_0)$
>
> In particular, when the initial state $s_0$ is fixed, an optimal policy that maximizes the worst-case state value (against the optimal state perturbation attack) can be found. Let $V^*(s_0)$ denote this optimal state value. Note that $V^*(s_0)$ is obtained using a different policy for different $s_0$. Then a reasonable definition of approximate Stackelberg Equilibrium is a policy $\pi$ where its state value under each state $s_0$ is close to $V^*(s_0)$, that is, $|V_{\pi\circ\omega_{\pi}}(s_0) - V^*(s_0)| \leq \Delta$ for some constant $\Delta$. Note that this condition should hold for each state $s_0$. Also note that we compare $\pi$ with a different optimal policy with respect to each $s_0$, thus bypassing the impossibility result.
>
> In the paper, we further approximated $V^*(s_0)$ by the optimal value without attacks and derived $\Delta$ in the simplified setting (Theorem 1). However, we believe that this definition presents a novel extension of approximate Stackelberg Equilibrium in the standard sense and will incorporate it into the next version of the paper.
>
> For Q4, we did not claim that our method can recover the exact true state from the perturbed state. Both our historical information-based and diffusion-based belief models generate a set of belief states $M_t$ to approximate the true state. Ideally, the belief states should provide a better approximation of the true state beyond what can be estimated from the epsilon ball centered at the perturbed state. Although we do not have a theoretical characterization of how the belief model reduces the defender’s uncertainty about the true state, we have provided some empirical evidence in the paper. Figures 5 (a)-(c) (Appendix F.3.2) show that the $l_2$ distance between the denoised state (generated from the belief) and the true state is significantly smaller than the $l_2$ distance between the perturbed state and the true state.
>
> Further, Theorem 1 shows that the performance loss of our pessimistic Q-learning approach (compared with the optimal policy without attacks) increases linearly over the defender’s uncertainty about the true state, showing the importance of reducing this uncertainty.
>
> We hope that our explanations address your concerns.

---

### Official Review · Reviewer_QHWx · 2023-11-01

**Soundness:** 3 good
**Presentation:** 3 good
**Contribution:** 4 excellent
**Rating:** 8
**Confidence:** 4

**Summary:**

This paper proposes two new algorithms to robustify agent policies against adversarial attacks. It formulates the problem of finding a robust policy as a Stacleberg game (where agents choose policies), and then further incorporates beliefs into the derived algorithm. For pixel-based observation spaces, the game uses a diffusion-based method to derive valid possible states. The paper is very well structured and easy to follow.

**Strengths:**

- the paper addresses many shortcomings of current works
- the theoretical algorithms and derivations are insightful
- the practical implementation of the derived algorithms are well motivated

**Weaknesses:**

- the paper assumes that both the clean MDP and the perturbation budget are known to both the victim and the attacker
- it would be interesting to run an ablation on these assumptions. How well does the method work if the budget is not known exactly, or if the MDP transition function is not known exactly?

**Questions:**

see weaknesses

---

> ### Author Response · Authors · 2023-11-16
>
> We appreciate your constructive comments and feedback. Now we will explain your concerns and questions point by point in the following.
>
> A1: How our methods perform when a clean environment is unavailable?
>
> Q1: We have added a general response to this question.
>
> A2: How our methods perform when the attack budget is unknown?
>
> Q2: Our BP-DQN method does not require accurate knowledge about the attack budget. The results in the paper were obtained using a single policy trained against the PGD attack with $\epsilon = 0.1$, without further adaptation at the test stage against the actual attack budget (0.1 or 0.5).
>
> Our DP-DQN methods also do not require accurate knowledge about the attack budget at the training stage. All the results shown in the paper for DP-DQN-O (resp. DP-DQN-F) were obtained using a single policy trained against the PGD attack with $\epsilon = 1/255$, and with the reverse steps $k= 5$ in DDPM (resp. $k=1$ in Progressive Distillation). However, we did adapt the value of $k$ at the test stage according to the actual attack budget in the paper. To understand the performance of our algorithms under an unknown attack budget at the test stage, we have conducted a new ablation study on these methods using the Pong environment with a fixed number of reverse steps in the diffusion model ($k = 30$ for DP-DQN-O and $k = 1$ for DP-DQN-F) and a fixed level of manually added noise (1/255) at the test stage, and tested them on PGD attacks with three different attack budgets (1/255, 3/255 and 15/255). The results for DP-DQN-O are $\epsilon = 1/255: 20 \pm 0$, $\epsilon = 3/255: 19.8 \pm 0.4$, $\epsilon = 15/255: 19.7 \pm 0.5$, which are on par with the results using an adapted $k$ shown in the paper. On the other hand, the results for DP-DQN-F are $\epsilon = 1/255: 20.4 \pm 0.9$, $\epsilon = 3/255: 13 \pm 7.4$, $\epsilon = 15/255: -20.6 \pm 0.4$, showing that it is more sensitive to attack budget. As a comparison, we have also tested other baselines using the policy trained under $\epsilon = 1/255$ against a PGD attack with budget $\epsilon = 3/255$, and the results are SA-DQN: $12 \pm 1.2$, WocaR-DQN: $-21 \pm 0$, showing that these methods are also sensitive to attack budget.
>
> We hope that our explanations address your major concerns. If there are still concerns or questions, we would be happy to hear and discuss them.

---

> > ### Comment · Reviewer_QHWx · 2023-11-21
> >
> > Dear authors, thank you for the additional clarifications and the additional experiments (which are very positive). I will maintain my score.

---

### Author Response · Authors · 2023-11-16
**General response for scenario when a clean environment is unavailable.**

As many reviewers asked how our methods will perform when a clean environment is unavailable, we answer this question in this general response.

We first note that existing studies (including all the baselines we used) on robust RL against adversarial state perturbations commonly assume that the agent has access to a clean environment at the training stage. This is mainly because an RL agent may suffer from a significant amount of loss if it has to explore a poisoned environment on the fly, which is infeasible for security-sensitive applications. However, we do agree that it is an important direction to explore and have conducted a preliminary investigation. We assumed that the agent could access a small amount of clean data (30 episodes of clean trajectories were used in the paper) to train a belief model or a diffusion model. Note that to learn the diffusion model, we only need a set of clean states but not necessarily complete trajectories. We then trained an RL policy in a poisoned environment by modifying the BP-DQN and DP-DQN algorithms as follows. For BP-DQN, we changed line 11 of Algorithm 4 (in Appendix E) to store the average of belief states (obtained from the perturbed states) instead of the true states in the replay buffer. For DP-DQN, we changed lines 8 and 9 of Algorithm 6 (in Appendix E) to apply the diffusion-based purification on perturbed states instead of true states.  We trained these policies against PGD-poisoned continuous Gridworld and Freeway environments, with an attack budget of 0.1 and 1/255, respectively, and then tested them in the same environment. The results are as follows.
|    Env  |   Model |  Natural Reward |         PGD        |         PGD        |         PGD         |       MinBest      |       MinBest      |       MinBest       |        PA-AD       |        PA-AD       |        PA-AD        |
|:-------:|:----------------------:|:----------------:|:------------------:|:------------------:|:-------------------:|:------------------:|:------------------:|:-------------------:|:------------------:|:------------------:|:-------------------:|
|         |          |                  | $\epsilon = 1/255$ | $\epsilon = 3/255$ | $\epsilon = 15/255$ | $\epsilon = 1/255$ | $\epsilon = 3/255$ | $\epsilon = 15/255$ | $\epsilon = 1/255$ | $\epsilon = 3/255$ | $\epsilon = 15/255$ |
| Freeway |  DP_DQN_F |  $29.4 \pm 0.9$  |   $29.0 \pm 1.2$   |   $28.8 \pm 1.5$   |    $29.0 \pm 1.4$   |   $28.8 \pm 0.4$   |   $29.6 \pm 1.1$   |     $29 \pm 1.6$    |   $29.2 \pm 0.4$   |   $28.2 \pm 1.8$   |    $23.6 \pm 1.1$   |

|           Env         |  Model |  Natural  Reward |        PGD       |        PGD       |      MinBest     |      MinBest     |       PA-AD      |       PA-AD      |
|:---------------------:|:------:|:----------------:|:----------------:|:----------------:|:----------------:|:----------------:|:----------------:|:----------------:|
|                       |        |                  | $\epsilon = 0.1$ | $\epsilon = 0.5$ | $\epsilon = 0.1$ | $\epsilon = 0.5$ | $\epsilon = 0.1$ | $\epsilon = 0.5$ |
| Continuous  Gridworld | BP_DQN |   $102 \pm 110$  |  $101.5 \pm 109$ |  $56.0 \pm 135$  |  $79.3 \pm 125$  |  $30.7 \pm 138$  |  $78.6 \pm 125$  |  $13.1 \pm 140$  |

Both our BP-DQN and DP-DQN-F methods suffer from performance loss brought by the poisoned environment compared with the results reported in the paper using a clean environment as expected. However, our methods still surpass all other baselines trained with a clean environment for continuous Gridworld as well as Freeway under large perturbations and strong attacks.

---

### Public Comment · ~Yongyuan_Liang1 · 2023-12-09
**Questions about experiments**

Hi Authors,

I find your experiments valuable in validating robustness under a very large attack epsilon—a scenario not explored in prior papers. However, I have two questions about your experiments:

1. In the experiments on Continuous Gridworld, SA-DQN and WocaR-DQN exhibit much worse performance than DQN on natural rewards and under a small attack budget. Considering that the only difference between SA-DQN and DQN is the state regularization, which has proven effective as a smoothness-inducing regularization across various tasks, I am curious why smooth policies training with the state regularization may have adverse effects in this scenario.

2. Additionally, why not compare your results with the robust baseline Radial-DQN (https://github.com/tuomaso/radial_rl)? Radial-DQN doesn't use state regularization but has limitations under a small attack budget. Including this baseline in your comparison could contribute to a more comprehensive evaluation.

Looking forward to following your work based on the release code!

---

### Meta-Review · Area_Chair_pmbc · 2023-12-08

**Metareview:**

The paper presents a robust RL algorithm to mitigate vulnerabilities to adversarial attacks. The paper's strengths and weaknesses, as identified by the reviewers, are summarized below:

Strengths:

+ Addressing Shortcomings of Current Works: The paper tackles many of the limitations in existing approaches to defend against adversarial attacks in RL.

+ Theoretical and Practical Insights: The theoretical algorithms and derivations provided are insightful and are well-supported by practical implementations. Although pessimistic Q-learning is similar to WocaR, belief state estimation and state purification are new.

Weaknesses:

- Computational Overhead: The addition of the diffusion model, while adding robustness, also increases the computational complexity, particularly at the test stage.

- Assumption of Clean Training Environment: The algorithm presupposes access to a clean training environment, which might not be feasible in many real-world scenarios.

- Non-Existence of Stackelberg Equilibrium: Concerns regarding the non-existence of Stackelberg equilibrium as defined in the paper, which could affect the validity of finding an approximate solution.

- Lack of Big Picture in Strategy Explanation: A need for a clearer overview of the strategies before delving into details, and the inclusion of more mathematical specifics in each section.

- Misrepresentation in Abstract: The abstract incorrectly states that past methods either regularize or retrain the policy, overlooking direct state purification methods.

- Incremental Innovations: Some innovations presented seem incremental, especially when compared to existing methods like WocaR-DQN. Although belief state estimation and state purification are new, they are computationally expensive.

**Justification For Why Not Higher Score:**

In addition to the weaknesses mentioned above, the author didn't seem to modify the manuscript during the rebuttal.

**Justification For Why Not Lower Score:**

Shown in the strengths above.

---

### Decision · Program_Chairs · 2024-01-16

Accept (poster)